# Finger somatotopy is preserved after tetraplegia but deteriorates over time

**Sanne Kikkert[1,2]\*, Dario Pfyffer[2], Michaela Verling[1], Patrick Freund[2,3,4,5], Nicole Wenderoth[1]**

[1]Neural Control of Movement Laboratory, Department of Health Sciences and Technology, ETH Zürich, Zürich, Switzerland; [2]Spinal Cord Injury Center, Balgrist University Hospital, University of Zürich, Zürich, Switzerland; [3]Department of Brain Repair and Rehabilitation, UCL Institute of Neurology, University College London, London, United Kingdom; [4]Wellcome Trust Centre for Neuroimaging, UCL Institute of Neurology, University College London, London, United Kingdom; [5]Department of Neurophysics, Max Planck Institute for Human Cognitive and Brain Sciences, Leipzig, Germany

**Abstract** Previous studies showed reorganised and/or altered activity in the primary sensorimotor cortex after a spinal cord injury (SCI), suggested to reflect abnormal processing. However, little is known about whether somatotopically specific representations can be activated despite reduced or absent afferent hand inputs. In this observational study, we used functional MRI and a (attempted) finger movement task in tetraplegic patients to characterise the somatotopic hand layout in primary somatosensory cortex. We further used structural MRI to assess spared spinal tissue bridges. We found that somatotopic hand representations can be activated through attempted finger movements in the absence of sensory and motor hand functioning, and no spared spinal tissue bridges. Such preserved hand somatotopy could be exploited by rehabilitation approaches that aim to establish new hand-brain functional connections after SCI (e.g. neuroprosthetics). However, over years since SCI the hand representation somatotopy deteriorated, suggesting that somatotopic hand representations are more easily targeted within the first years after SCI.

**\*For correspondence:**
sanne.kikkert@hest.ethz.ch

**Competing interest:** The authors declare that no competing interests exist.

## Introduction

A spinal cord injury (SCI) refers to damage of the spinal cord caused by a trauma, disease, or degeneration of vertebral discs. SCI mostly results in a partial or complete loss of motor control and sensation. Following a tetraplegia (or cervical SCI), individuals mostly experience a loss of muscle function and sensation in their limbs and torso (*Curt et al., 1998*; *Kalsi-Ryan et al., 2014*; *Petersen et al., 2012*). Accordingly, the primary somatosensory cortex (S1) mostly receives weakened, or is fully deprived of, afferent inputs and is exposed to altered motor behaviour (*Ozdemir and Perez, 2018*).

Research in non-human primate models of chronic and complete cervical SCI has shown that the S1 hand area becomes largely unresponsive to tactile hand stimulation after the injury (*Jain et al., 2008*; *Kambi et al., 2014*; *Liao et al., 2021*). The surviving finger-related activity became disorganised such that a few somatotopically appropriate sites but also other somatotopically non-matched sites were activated (*Liao et al., 2021*). Seminal non-human primate research has further demonstrated that SCI leads to extensive cortical reorganisation in S1, such that tactile stimulation of cortically adjacent body parts (e.g. of the face) activated the deprived brain territory (e.g. of the hand; *Halder et al., 2018*; *Jain et al., 2008*; *Kambi et al., 2014*). Although the physiological hand representation appears to largely be altered following a chronic cervical SCI in non-human primates, the anatomical isomorphs of individual fingers are unchanged (*Jain et al., 1998*). This suggests that while a hand representation

can no longer be activated through tactile stimulation after the disruption of afferent spinal pathways, a latent and somatotopic hand representation may be preserved regardless of large-scale physiological reorganisation.

A similar pattern of results has been reported for human SCI patients. Transcranial magnetic stimulation (TMS) studies induced current in localised areas of SCI patients' primary motor cortex (M1) to induce a peripheral muscle response. They found that representations of more impaired muscles retract or are absent while representations of less impaired muscles shift and expand (*Fassett et al., 2018*; *Freund et al., 2011a*; *Levy et al., 1990*; *Streletz et al., 1995*; *Topka et al., 1991*; *Urbin et al., 2019*). Similarly, human fMRI studies have shown that cortically neighbouring body part representations can shift towards, though do not invade, the deprived M1 and S1 cortex (*Freund et al., 2011b*; *Henderson et al., 2011*; *Jutzeler et al., 2015*; *Wrigley et al., 2018*; *Wrigley et al., 2009*). Other human fMRI studies hint at the possibility of latent somatotopic hand representations following SCI by showing that attempted movements with the paralysed and sensory-deprived body part can still evoke signals in the sensorimotor system (*Cramer et al., 2005*; *Freund et al., 2011b*; *Kokotilo et al., 2009*; *Solstrand Dahlberg et al., 2018*). This attempted 'net' movement activity was, however, shown to substantially differ from healthy controls: activity levels have been shown to be increased (*Freund et al., 2011b*; *Kokotilo et al., 2009*; *Solstrand Dahlberg et al., 2018*) or decreased (*Hotz-Boendermaker et al., 2008*), volumes of activation have been shown to be reduced (*Cramer et al., 2005*; *Hotz-Boendermaker et al., 2008*), activation was found in somatotopically non-matched cortical sites (*Freund et al., 2011b*), and activation was poorly modulated when patients switched from attempted to imagined movements (*Cramer et al., 2005*). These observations have therefore mostly been attributed to abnormal and/or disorganised processing induced by the SCI. It remains possible though that, despite certain aspects of sensorimotor activity being altered after SCI, somatotopically typical representations of the paralysed and sensory deprived body parts can be preserved (e.g. finger somatotopy of the affected hand(s)). Such preserved representations have the potential to be exploited in a functionally meaningful manner (e.g. via neuroprosthetics).

Case studies using intracortical stimulation in the S1 hand area to elicit finger sensations in tetraplegic patients hint at such preserved somatotopic representations (*Fifer et al., 2020*; *Flesher et al., 2016*), with one exception (*Armenta Salas et al., 2018*). Negative results were suggested to be due to a loss of hand somatotopy and/or reorganisation in S1 of the implanted tetraplegic patient, or due to potential misplacement of the implant (*Armenta Salas et al., 2018*). Whether fine-grained somatotopy is generally preserved in the tetraplegic patient population remains unknown. It is also unclear what clinical, behavioural, and structural spinal cord determinants may influence such representations to be maintained.

Here we used functional MRI (fMRI) and a visually cued (attempted) finger movement task in tetraplegic patients to examine whether hand somatotopy is preserved following a disconnect between the brain and the periphery. We instructed patients to perform the fMRI tasks with their most impaired upper limb and matched control participants' tested hands to patients' tested hands. If a patient was unable to make overt finger movements due to their injury, then we carefully instructed them to make attempted (i.e. not imagined) finger movements. To see whether patient's maps exhibited characteristics of somatotopy, we visualised finger selectivity in S1 using a travelling wave approach. To investigate whether fine-grained hand somatotopy was preserved and could be activated in S1 following tetraplegia, we assessed inter-finger representational distance patterns using representational similarity analysis (RSA). These inter-finger distance patterns are thought to be shaped by daily life experience such that fingers used more frequently together in daily life have lower representational distances (*Ejaz et al., 2015*). RSA-based inter-finger distance patterns have been shown to depict the invariant representational structure of fingers in S1 and M1 better than the size, shape, and exact location of the areas activated by finger movements (*Ejaz et al., 2015*). Over the past years RSA has therefore regularly been used to investigate somatotopy of finger representations both in healthy (e.g. *Akselrod et al., 2017*; *Ariani et al., 2020*; *Ejaz et al., 2015*; *Gooijers et al., 2021*; *Kieliba et al., 2021*; *Kolasinski et al., 2016b*; *Liu et al., 2021*; *Sanders et al., 2019*) and patient populations (e.g. *Dempsey-Jones et al., 2019*; *Ejaz et al., 2016*; *Kikkert et al., 2016*; *Wesselink et al., 2019*). We closely followed procedures that have previously been used to map preserved and typical somatotopic finger selectivity and inter-finger representational distance patterns of amputees' missing hands in S1 using volitional phantom finger movements (*Kikkert et al., 2016*; *Wesselink et al., 2019*).

However, in amputees, these movements generally recruit the residual arm muscles that used to control the missing limb via intact connections between the brain and spinal cord. Whether similar preserved somatotopic mapping can be observed in SCI patients with diminished or no connections between the brain and the periphery is unclear. If finger somatotopy is preserved in tetraplegic patients, then we should find typical inter-finger representational distance patterns in the S1 hand area of these patients. By measuring a group of 14 chronic tetraplegic patients with varying amounts of spared spinal cord tissue at the lesion level (quantified by means of midsagittal tissue bridges based on sagittal T2w scans), we uniquely assessed whether preserved connections between the brain and periphery are necessary to preserve fine somatotopic mapping in S1 (*Huber et al., 2017*; *Pfyffer et al., 2019*). If spared connections between the periphery and the brain are not necessary for preserving hand somatotopy, then we should find typical inter-finger representational distance patterns even in patients without spared spinal tissue bridges. We also investigated what clinical and behavioural determinants may contribute to preserving S1 hand somatotopy after chronic tetraplegia. If spared sensorimotor hand function is not necessary for preserving hand somatotopy, then we should find typical inter-finger representational distance patterns even in patients who suffer from full sensory loss and paralysis of the hand(s).

## Results

### Patient impairments

We tested 14 chronic tetraplegic patients that were heterogenous in terms of completeness of the SCI (ranging from AIS-A to AIS-D), neurological level of the injury (ranging from C2 to C7), years since injury (ranging from 6 months to 33 years since SCI), and sensorimotor upper limb impairments (ranging from a Graded Redefined Assessment of Strength, Sensibility and Prehension test (GRASSP) score for both limbs of 21 to 220; healthy GRASSP score = 232). We further tested 18 age-, sex-, and handedness-matched healthy control participants.

### Finger selectivity is preserved following tetraplegia

We used 3T fMRI and a travelling wave paradigm to investigate the somatotopic layout of finger selectivity on the S1 cortical surface (*Besle et al., 2013*; *Kolasinski et al., 2016a*). To visualise whether there was a consistent and somatotopic layout of finger selectivity across participants in the control and SCI patient groups, we created probability maps of finger selectivity. A characteristic hand map

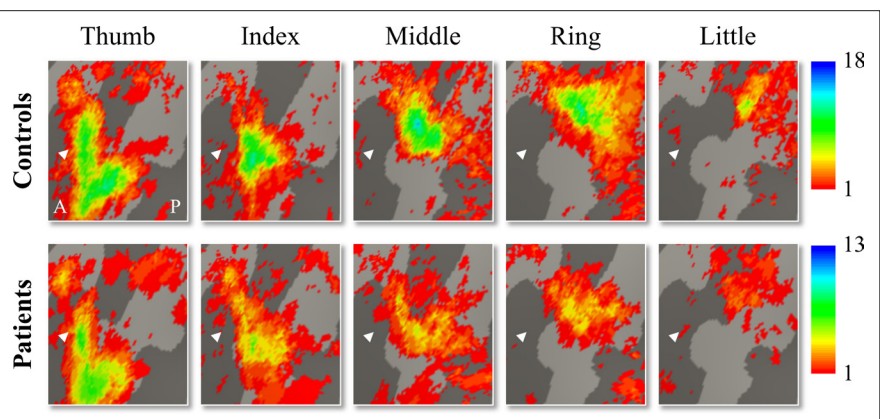

**Figure 1.** Inter-participant somatotopic finger-specific probability maps of the control and tetraplegic patient groups. Colours indicate the number of participants (ranging from 1 [red] till 18 and 13 [blue] participants for the control and SCI patient group, respectively) who demonstrated finger selectivity for a given vertex. Characteristic finger selectivity is characterised by a progression of finger selectivity from the thumb (laterally) to the little finger (medially). These characteristic finger progressions can be observed in both the control (top) and the tetraplegic patient (bottom) group's probability maps. Qualitative inspection suggests that inter-participant consistency was lowest for the little finger representation in both groups. It further appears that overall inter-participant consistency was reduced in the patient group compared to the control group. White arrows indicate the central sulcus. A: anterior; P: posterior.

shows a gradient of finger preference, progressing from the thumb (laterally) to the little finger (medially). We found a characteristic progression of finger selective clusters in both the control and tetraplegic patient groups (*Figure 1*). Qualitative inspection suggests that inter-participant consistency was lowest for the little finger representation in both groups. It further appears that overall inter-participant consistency was reduced in the patient group compared to the control group.

Given the known the relatively high inter-participant variability in finger selectivity (*Kolasinski et al., 2016a*) and the clinical heterogeneity of our SCI patient group, we further visualised each individual participant's finger selectivity map (*Figure 2A and B*). Overall, we found aspects of somatotopic finger selectivity in the maps of SCI patients' hands, in which neighbouring clusters showed selectivity for neighbouring fingers in contralateral S1, similar to those observed in 18 age-, sex-, and handedness-matched healthy controls. Notably, a characteristic hand map was even found in a patient who suffered complete paralysis and sensory deprivation of the hands (*Figure 2B*, patient map 1; patient S01). Despite most maps (*Figure 2*, except patient map 3; patient S04) displaying aspects of characteristic finger selectivity, some finger representations were not visible in the thresholded patient and control maps.

To ensure that the observed finger selective clusters were not representing noise, but rather true finger selectivity, we calculated split-half consistency between two halves of the travelling wave dataset using the Dice overlap coefficient (DOC; *Dice, 1945*). Minimally thresholded finger maps were compared across the split-halves of the data within an S1 mask. Overall, split-half consistency was not significantly different between patients and controls, as tested using a robust mixed ANOVA (see *Figure 2C*; $F_{(1,20.32)} = 0.51$, p=0.48). There was a significant difference in split-half consistency between pairs of same, neighbouring, and non-neighbouring fingers ($F_{(2,18.59)} = 159.69$, p<0.001). This neighbourhood relationship was not significantly different between the control and patient groups (i.e. there was no significant interaction; $F_{(2,18.59)} = 2.44$, p=0.11).

The DOC was highest for comparison of the same fingers between two halves of the dataset compared to neighbouring (controls: W = 171, p<0.001, $BF_{10} = 2761.24$; patients: W = 91, p<0.001, $BF_{10} = 742.45$) and non-neighbouring fingers (controls: W = 171, p<0.001, $BF_{10} = 248.14$; patients: W = 91, p<0.001, $BF_{10} = 133.05$). Moreover, neighbouring fingers showed greater overlap across the split-halves of the dataset than non-neighbouring fingers (controls: W = 171, p<0.001, $BF_{10} = 835.29$; patients: W = 91, p=0.001, $BF_{10} = 65.32$). This demonstrates that there was a somatotopic gradient in split-half consistency that was similar between the control and patient groups, demonstrating that the finger maps represented true finger selectivity.

## Typical hand somatotopy is preserved following tetraplegia

Next, we assessed 3T univariate task-related activity during individual finger movements performed in a blocked design fashion. Task-related activity was quantified by extracting the percent signal change for finger movement (across all fingers) versus baseline within the contralateral S1 hand area (see *Figure 3A*). Overall, all patients were able to engage their S1 hand area by moving individual fingers ($t_{(13)} = 7.46$, p<0.001; $BF_{10} = 4.28 e + 3$), as did controls ($t_{(17)} = 9.92$, p<0.001; $BF_{10} = 7.40 e + 5$). Furthermore, patients' task-related activity was not significantly different from controls ($t_{(30)} = –0.82$, p=0.42; $BF_{10} = 0.44$), with the Bayes factor (BF) showing anecdotal evidence in favour of the null hypothesis. Similar results were found when exploring univariate task-related activity in the contralateral M1 hand ROI (see *Figure 3—figure supplement 1*).

While the travelling wave maps demonstrate finger selectivity, they provide little information about the overlap between finger representations. We therefore examined the intricate relationship between finger representations in the S1 hand area for all patients and controls using RSA (see *Figure 3B and C*). The resulting inter-finger distances were averaged across finger pairs within each participant to obtain an estimate for average inter-finger separability (see *Figure 3D*). We found that inter-finger separability in the S1 hand area was greater than 0 for patients ($t_{(13)} = 9.83$, p<0.001; $BF_{10} = 6.77 e + 4$) and controls ($t_{(17)} = 11.70$, p<0.001; $BF_{10} = 6.92 e + 6$), indicating that the S1 hand area in both groups contained information about individuated finger representations. Furthermore, for both controls (W = 171, p<0.001; $BF_{10} = 4059$) and patients (W = 105, p<0.001; $BF_{10} = 279$) there was significant greater separability (or representation strength) in the S1 hand area than in a control cerebral spinal fluid (CSF) ROI that would not be expected to contain information about individuated finger representations. We did not find a significant group difference in inter-finger separability of

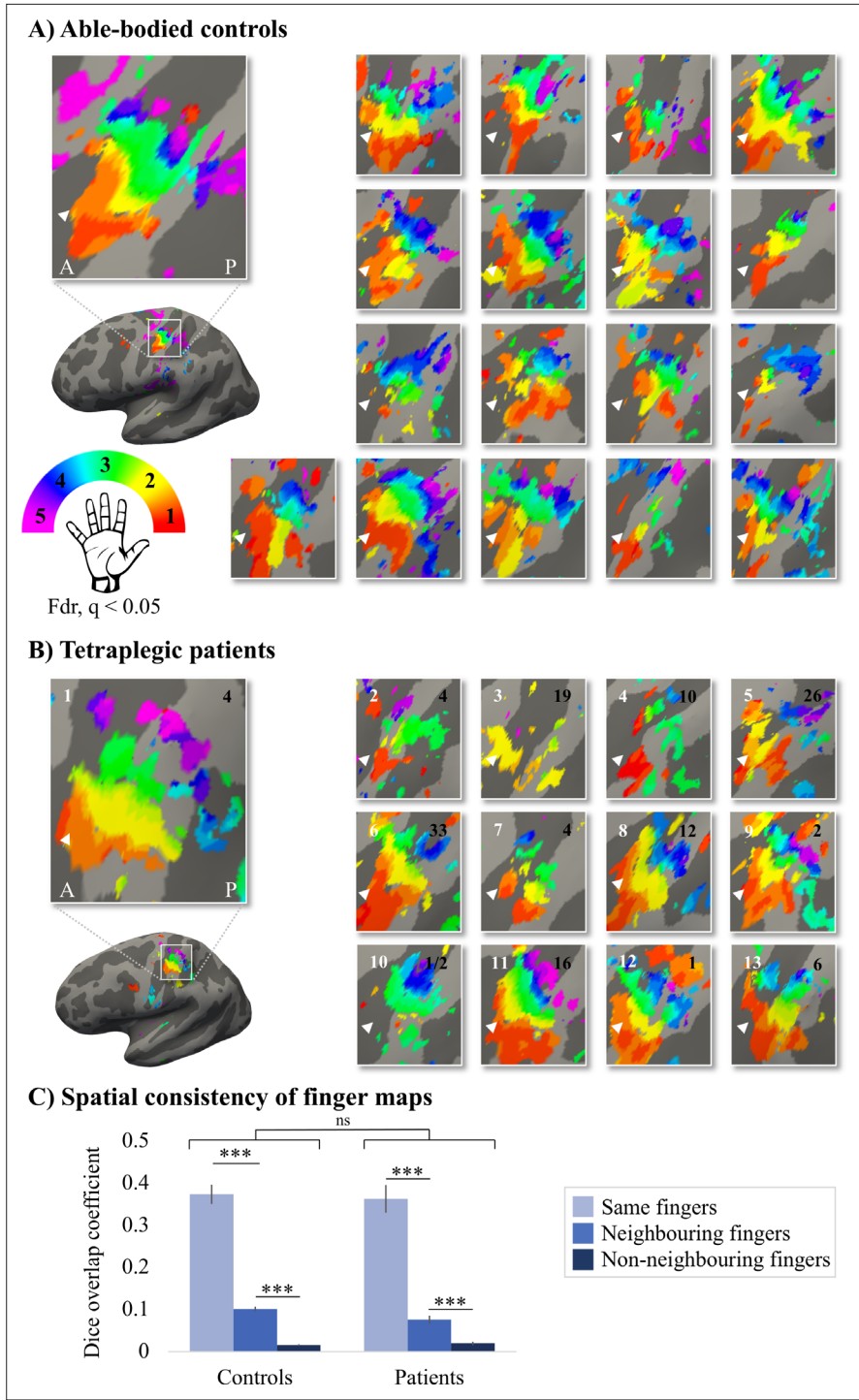

**Figure 2.** Finger selectivity is preserved in tetraplegic patients. Colours indicate selectivity for the thumb (finger 1, red), index finger (finger 2, yellow), middle finger (finger 3, green), ring finger (finger 4, blue), and little finger (finger 5, purple). Maps of participants for whom the left hand was tested are horizontally mirrored for visualisation purposes. Typical finger selectivity is characterised by a gradient of finger preference, progressing from the thumb (laterally) to the little finger (medially). These characteristic gradients of finger selectivity can be observed in both the able-bodied controls (**A**) and the tetraplegic patients (**B**). Despite most maps (except patient map 3) displaying aspects of characteristic finger maps, some finger representations were not visible in the thresholded patient and control maps. Patients' hand maps are sorted according to their overall upper-limb impairments (assessed using the Graded Redefined Assessment of Strength, Sensibility and Prehension test [GRASSP]): from most to least impaired – as indicated by the white numbers. Black numbers indicate the years since spinal cord

*Figure 2 continued on next page*

*Figure 2 continued*

injury (SCI). Multiple comparisons were adjusted using a false discovery rate (FDR) with q < 0.05. Other figure annotations are as in *Figure 1*. (**C**) To ensure that the observed clusters were not representing noise, but rather true finger selectivity, we calculated split-half consistency between two halves of the minimally thresholded (Z > 2) travelling wave dataset (see *Figure 2—figure supplement 1* for the travelling wave maps used to calculate split-half consistency). Both controls and patients showed higher split-half consistency (assessed using the Dice overlap coefficient) for comparison of the same fingers between two halves of the travelling wave dataset (light blue), compared to neighbouring (blue), and non-neighbouring fingers (dark blue). Moreover, neighbouring fingers showed greater overlap across the split-halves of the dataset then non-neighbouring fingers for both patients and controls. The same results were obtained when calculating split-half consistency on maps thresholded using FDR q < 0.05 (as was used for the maps in **A, B**; see *Figure 2—figure supplement 2*). Error bars show the standard error of the mean. *** = corrected p≤0.001, ns: non-significant.

The online version of this article includes the following figure supplement(s) for figure 2:

**Figure supplement 1.** Hard-edged split-half travelling wave maps used to calculate the intra-participant spatial consistency reported in *Figure 2C*.

**Figure supplement 2.** Spatial consistency of false discovery rate (FDR)-thresholded finger maps.

the S1 hand area ($t_{(30)}$ = 1.52, p=0.14; $BF_{10}$ = 0.81), with the BF showing anecdotal evidence in favour of the null hypothesis.

We then tested whether the inter-finger distances were different across finger pairs between controls and tetraplegic patients using a robust mixed ANOVA with a within-participants factor for finger pair (10 levels) and a between-participants factor for group (two levels: controls and tetraplegic patients; *Figure 3—figure supplement 2*). We did not find a significant difference in inter-finger distances between patients and controls ($F_{(1,21.66)}$ = 1.50, p=0.23). The inter-finger distances were significantly different across finger pairs, as would be expected based on somatotopic mapping ($F_{(9,15.38)}$ = 27.22, p<0.001). This pattern of inter-finger distances was not significantly different between groups (i.e. no significant finger pair by group interaction; $F_{(9,15.38)}$ = 1.05, p=0.45). When testing for group differences per finger pair, the BF only revealed inconclusive evidence (BF >0.37 and < 1.11; note that we could not run a Bayesian ANOVA due to normality violations).

Although inter-finger separability was not significantly different between patients and controls, it is possible that the pattern of inter-finger distances was atypical in the patients. We therefore examined whether the inter-finger distance pattern was normal (or typical) in tetraplegic patients (see *Figure 3E*) by correlating each participant's inter-finger distance pattern with a canonical inter-finger distance pattern. Tetraplegic patients' typicality scores were compared to those of the controls and of a group congenital one-handers (data taken from an independent study; *Wesselink et al., 2019*). Congenital one-handers are born without a hand and therefore do not have a cortical representation of the missing hand (unlike amputees who develop the representation before losing the hand and hence have a 'missing hand representation'; *Wesselink et al., 2019*). This group was therefore included as a control for absence of hand representation. We found a significant difference in typicality between tetraplegic patients, healthy controls, and congenital one-handers ($H_{(2)}$ = 26.64, p<0.001). As expected, post hoc tests revealed significantly higher typicality in controls compared to congenital one-handers (U = 0, p<0.001; $BF_{10}$ = 113.60). Importantly, inter-finger distances typicality of the SCI patients was significantly higher than the typicality scores of the congenital one-handers (U = 4, p<0.001; $BF_{10}$ = 90.33), but not significantly different from the typicality scores of the controls (U = 103, p=0.40; $BF_{10}$ = 0.55). The BF for the comparison between SCI patients and controls showed anecdotal evidence for equivalence between both groups.

## Typical hand somatotopy deteriorates over years after tetraplegia

Next, we aimed to understand which clinical, behavioural, and structural spinal cord determinants may allow hand representations in S1 to be maintained. We first explored correlations with S1 hand representation typicality. We found that the number of years since SCI significantly correlated with hand representation typicality (see *Figure 4A*; $r_s$ = –0.59, p=0.028), suggesting that S1 hand representation typicality may deteriorate over time after SCI. We further found that patients with more retained GRASSP motor function of the tested upper limb had more typical hand representations in S1 (see *Figure 4B*; $r_s$ = 0.60, p=0.02). We did not find a significant correlation between S1 hand

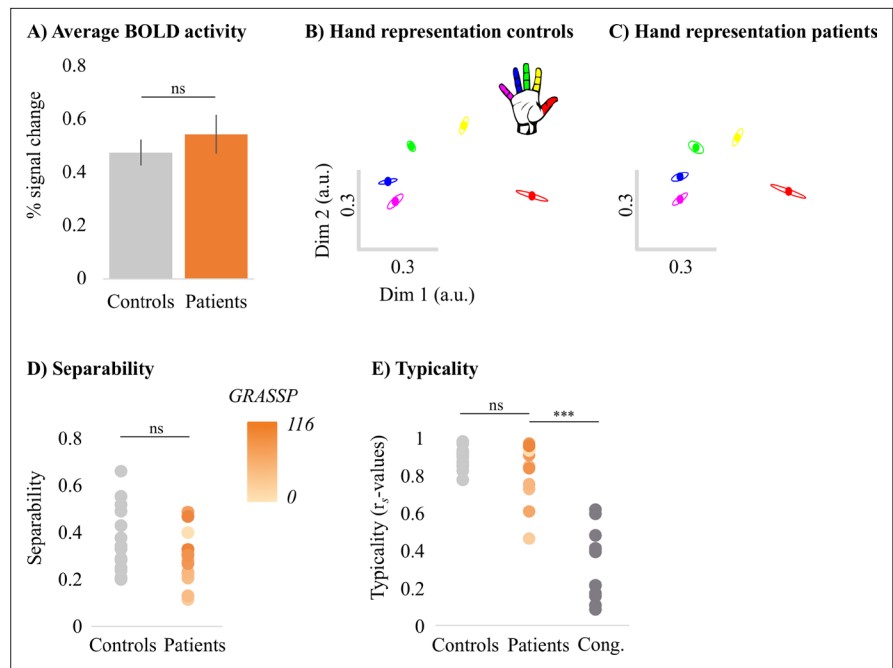

**Figure 3.** Typical multivariate hand somatotopy is preserved following tetraplegia. (**A**) Percent signal change in the S1 hand area during finger movement for able-bodied controls (grey) and tetraplegic patients (orange). Similar results were found in the M1 hand ROI (see *Figure 3—figure supplement 1*). (**B, C**) Two-dimensional projection of the representational structure of inter-finger distances in the control (**B**) and tetraplegic patient groups (**C**). Inter-finger distance is reflected by the distance in the two dimensions. Individual fingers are represented by different colours: thumb, red; index finger, yellow; middle finger, green; ring finger, blue; little finger, purple. Ellipses represent the between-participants' standard error after Procrustes alignment. Inter-finger distances across finger pairs were significantly different across finger pairs (as would be expected based on somatotopic mapping), but not between controls and tetraplegic patients (see *Figure 3—figure supplement 2*). Individual participant inter-finger distance patterns are visualised in *Figure 3—figure supplement 3* and *Figure 3—figure supplement 4* for the controls and patients, respectively. (**D**) Separability, measured as mean inter-finger distance, of the representational structure in the S1 hand area of controls and patients. Patients are presented on a colour scale representing the sensory and motor functioning of their tested upper limb, measured using the Graded Redefined Assessment of Strength, Sensibility and Prehension test (GRASSP) (0 = no upper limb function, 116 = normal upper limb function). (**E**) Typicality of the representational structure in controls, patients, and congenital one-handers (Cong. in the figure). *** p<0.001; ns: non-significant; Dim: dimension; a.u.: arbitrary unit; Cong: congenital one-handers.

The online version of this article includes the following figure supplement(s) for figure 3:

**Figure supplement 1.** Percent signal change in the M1 hand area during finger movement.

**Figure supplement 2.** Inter-finger distances across finger pairs for controls and tetraplegic patients.

**Figure supplement 3.** Individual control participant's inter-finger distance patterns.

**Figure supplement 4.** Individual tetraplegic patient's inter-finger distance patterns.

representation typicality and GRASSP sensory function of the tested upper limb, spared midsagittal spinal tissue bridges at the lesion level, or cross-sectional spinal cord area (see *Figure 4C–E*; $r_s$ = 0.40, p=0.16, $r_s$ = 0.46, p=0.14, and $r_s$ = 0.33, p=0.25, respectively).

We further explored the hand representation typicality of patients S01 and S03 who did not have any spared midsagittal spinal tissue bridges at the lesion level, a complete (S01) or near complete (S03) hand paralysis, and a complete (S01) or near complete loss (S03) of hand sensory function (as assessed using the GRASSP test). Interestingly, both patients had a highly typical hand representation in S1 that was significantly different from congenital one-handers (i.e. who are born without a hand and do not have a missing hand representation; S01: $t_{(12)}$ = 3.20, p=0.008; S03: $t_{(12)}$ = 2.97, $P$ = 0.01), but not controls (S01: $t_{(17)} =$ 0.95, p=0.36; S03: $t_{(17)}$ = 0.04, p=0.97). This suggests that retained

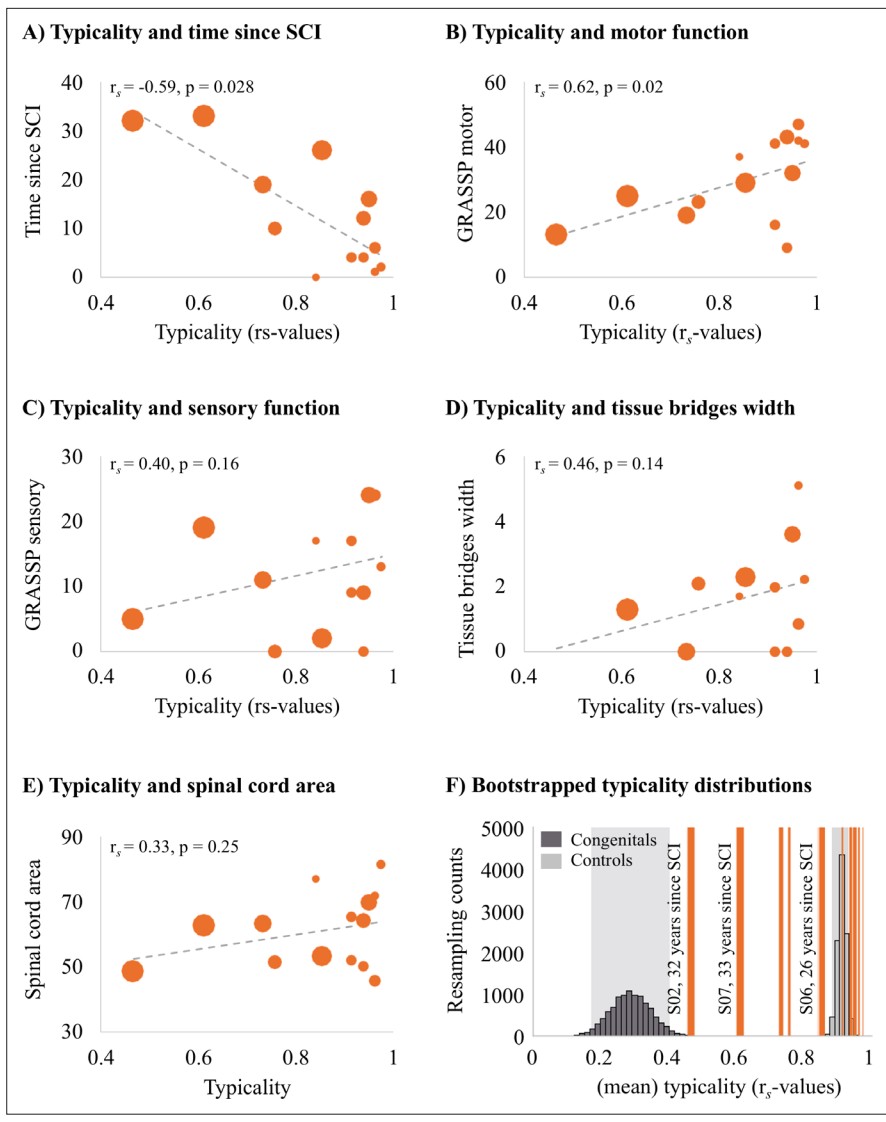

**Figure 4.** Years since spinal cord injury and retained motor function correlate with hand representation typicality in the primary somatosensory cortex (S1). We examined clinical, behavioural, and spinal cord structural correlates for hand representation typicality. Increasing marker sizes represent increasing years since spinal cord injury (SCI) in graphs **A–E**. (**A**) There was a negative correlation between years since SCI and hand representation typicality. (**B**) We found a positive correlation between motor function of the tested upper limb (measured using the Graded Redefined Assessment of Strength, Sensibility and Prehension test [GRASSP]) and hand representation typicality. There was no significant correlation between hand representation typicality and sensory function of the tested upper limb (**C**; measured using the GRASSP), spared midsagittal spinal tissue bridges (**D**), and cross-sectional spinal cord area (**E**). (**F**) Bootstrapped distribution of controls' and congenital one-handers' mean S1 hand representation typicality. Dark grey bars indicate the distribution of congenital one-handers (data taken from an independent study; **Wesselink et al., 2019**), and light grey bars indicate the distribution of the able-bodied controls (tested for this study). The typicality scores of the SCI patients are plotted as orange lines. Increasing line thickness represent increasing years since SCI. Grey shaded areas indicate the 95% confidence intervals of the mean for congenital one-handers and able-bodied controls.

connections between the periphery and the brain, retained motor functioning, and retained sensory functioning may not be necessary to maintain typical hand representations in S1.

We then ran an exploratory stepwise linear regression to investigate which of these clinical, behavioural, and structural spinal cord characteristics were predictive of hand representation typicality in S1. Years since SCI significantly predicted hand representation typicality in S1 with $R^2 = 0.40$ ($F_{(1,10)} = 6.73$, p=0.027). Motor function of the tested upper limb, sensory function of the tested hand, spared

midsagittal spinal tissue bridges at the lesion level, and spinal cord area did not significantly add to the prediction (t = 1.43, p=0.19, t = 1.44, p=0.18, t = 1.19, p=0.26, and t = 0.41, p=0.69, respectively). This analysis suggests that while hand representations are preserved following tetraplegia, they may deteriorate over time.

To inspect this further, we bootstrapped the mean typicality of the able-bodied controls and congenital one-handers 10,000 times to infer their population means. While most tetraplegic patients' typicality scores fell in, or very close to, the able-bodied controls' distribution, we found that some SCI patients' typicality scores fell in-between the able-bodied controls' and congenital one-handers' distributions (*Figure 4F*). This suggests that these patients may not be similar to congenital one-handers or to able-bodied controls. Interestingly, this included those patients for whom most years had passed since their SCI. This suggests that S1 hand representations might deteriorate after an SCI, but some weak hand information may be maintained in S1 even >30 years after an SCI.

## Discussion

In this study, we investigated whether hand somatotopy is preserved and can be activated through attempted movements following tetraplegia. We tested a heterogenous group of SCI patients to examine what clinical, behavioural, and structural spinal cord determinants contribute to preserving S1 somatotopy. Our results revealed that detailed hand somatotopy can be preserved following tetraplegia, even in the absence of sensory and motor hand function and a lack of spared spinal tissue bridges. However, over time since SCI these finger maps deteriorated such that the hand somatotopy became less typical.

Spared spinal cord tissue bridges can be found in most patients with a clinically incomplete injury, their width being predictive of electrophysiological information flow, recovery of sensorimotor function, and neuropathic pain (*Huber et al., 2017*; *Pfyffer et al., 2021*; *Pfyffer et al., 2019*; *Vallotton et al., 2019*). However, in this study, spared midsagittal spinal tissue bridges at the lesion level and sensorimotor hand function did not seem necessary to maintain and activate a somatotopic hand representation in S1. We found a highly typical hand representation in two patients (S01 and S03) who did not have any spared spinal tissue bridges at the lesion level, a complete (S01) or near complete (S03) hand paralysis, and a complete (S01) or near complete loss (S03) of hand sensory function. Our predictive modelling results were in line with this notion and showed that these behavioural and structural spinal cord determinants were not predictive of hand representation typicality. Note however that our sample size was limited, and it is challenging to draw definite conclusions from non-significant predictive modelling results.

Time since injury was predictive of a deteriorated, or less typical, somatotopic S1 hand representation. In fact, both patients with a typical S1 hand representations but absent spinal tissue bridges suffered their SCI only 4 years ago (the group was on average 12 years since SCI). The hand representation typicality of patients who suffered their SCI further in the past were not similar to congenital one-handers' nor to controls' hand representation. Thus, S1 hand representations may deteriorate over time after an SCI, but some weak hand information appears to be maintained in S1 even >30 years after an SCI. This finding complements previous studies in amputees showing that missing hand somatotopy is preserved even decades after arm amputation and that years since injury was not related to missing hand representation typicality (*Kikkert et al., 2016*; *Wesselink et al., 2019*). While both amputees and SCI patients suffer from major sensory input loss and changed motor behaviour, their injuries are inherently different. The injured axons within amputees' residual limb and the remaining part of the peripheral nerves mostly generate some spontaneous (ectopic) activity that is propagated to the brain and could contribute to maintaining S1 representations (*Kikkert et al., 2019*; *Kikkert et al., 2018*; *Kikkert et al., 2016*; *Makin et al., 2013*; *Nyström and Hagbarth, 1981*; *Vaso et al., 2014*). Furthermore, amputees mostly remain able to move and receive afferents from the residual limb muscles that used to control the missing hand as most of their motor system remains intact. Although their hand is missing, amputees' residual arm muscles are often still used (either to move the residual limb and/or to control a prosthetic arm). Lastly, amputees' vividness of kinaesthetic sensations during phantom finger movements was found to be predictive of the typicality of the S1 missing hand representation (*Wesselink et al., 2019*). A continued, though altered, experience relating to the missing hand in amputees may contribute to maintaining the somatotopic missing hand representation in S1. Contrarily, tetraplegic patients mostly have reduced or a complete loss

of communication between the brain and periphery. They therefore have problems activating the adequate muscles and will lose orderly afferents from their muscles and the skin. It is possible that this continued disuse causes somatotopic S1 representations to deteriorate after SCI.

How may these representations be preserved over time and activated through attempted movements in the absence of peripheral information? S1 is reciprocally connected with various brain areas, for example,, M1, lateral parietal cortex, posterior parietal area 5, secondary somatosensory cortex, and supplementary motor cortex (*Delhaye et al., 2018*). After a loss of sensory inputs and paralysis through SCI, S1 representations may be activated and preserved through its interconnections with these areas. Firstly, it is possible that cortico-cortical efference copies may keep a representation 'alive' through occasional corollary discharge (*London and Miller, 2013*). While motor and sensory signals no longer pass through the spinal cord in the absence of spinal tissue bridges, S1 and M1 remain intact. When a motor command is initiated (e.g. in the form of an attempted hand movement), an efference copy is thought to be sent to S1 in the form of corollary discharge. This corollary discharge resembles the expected somatosensory feedback activity pattern and may drive somatotopic S1 activity even in the absence of ascending afferent signals from the hand (*Adams et al., 2013*; *London and Miller, 2013*). It is possible that our patients occasionally performed attempted movements which would result in corollary discharge in S1. Second, it is likely that attempting individual finger movements poses high attentional demands on tetraplegic patients. Accordingly, attentional processes might have contributed to eliciting somatotopic S1 activity. Evidence for this account comes from studies showing that it is possible to activate somatotopic S1 hand representations through attending to individual fingers (*Puckett et al., 2017*) or through touch observation (*Kuehn et al., 2018*). Attending to fingers during our attempted finger movement task may have been sufficient to elicit somatotopic S1 activity through top-down processes in the tetraplegic patients who lacked motor and sensory hand function. Furthermore, one might speculate that observing others' or one's own fingers being touched or directing attention to others' hand movements or one's own fingers may help preserve somatotopic representations. Third, it is possible that these somatotopic maps are relatively hardwired, and while they deteriorate over time, they never fully disappear. Indeed, somatotopic mapping of a sensory-deprived body part has been shown to be resilient after dystonia (*Ejaz et al., 2016*; though see *Burman et al., 2008* and *Elbert et al., 1998*) and arm amputation (*Bruurmijn et al., 2017*; *Kikkert et al., 2016*; *Wesselink et al., 2019*). Fourth, it is possible that even though a patient is clinically assessed to be complete and is unable to perceive sensory stimuli on the deprived body part, there is still some ascending information flow that contributes to preserving somatotopy (*Wrigley et al., 2018*). A recent study found that although complete paraplegic SCI patients were unable to perceive a brushing stimulus on their toe, 48% of patients activated the location-appropriate S1 area (*Wrigley et al., 2018*). However, the authors of this study defined the completeness of patients' injuries via behavioural testing, while we additionally assessed the retained connections passing through the SCI directly via quantification of spared spinal tissue bridges through structural MRI. It is unlikely that spinal tissue carrying somatotopically organised information would be missed by our assessment (*Huber et al., 2017*; *Pfyffer et al., 2019*). Our experiment did not allow us to tease apart these potential processes, and it is likely that various processes simultaneously influence the preservation of S1 somatotopy and elicited the observed somatotopic S1 activity.

Our finding of preserved S1 somatotopy may appear inconsistent with the wealth of evidence showing cortical reorganisation in S1 following SCI (*Halder et al., 2018*; *Jain et al., 2008*; *Kambi et al., 2014*). In these studies, experimenters indirectly probed the deprived S1 hand cortex via stimulation of cortically adjacent body parts. Human fMRI studies similarly probed the intact and cortically neighbouring body parts and suggested that their representations shift towards the deprived S1 cortex (*Freund et al., 2011b*; *Freund et al., 2011a*; *Henderson et al., 2011*; *Jutzeler et al., 2015*; *Wrigley et al., 2018*; *Wrigley et al., 2009*). TMS studies similarly induce current in localised areas of M1 to induce a peripheral muscle response of cortically neighbouring body parts. These studies demonstrated that the representations of less impaired muscles shift and expand following a complete or incomplete SCI (*Fassett et al., 2018*; *Freund et al., 2011a*; *Levy et al., 1990*; *Streletz et al., 1995*; *Topka et al., 1991*; *Urbin et al., 2019*). Our fMRI results showed that tetraplegic patients had a preserved somatotopic hand representation in S1, though this deteriorated over time. We did not probe body parts other than the hand and could therefore not investigate whether any remapping of other (neighbouring and/or intact) body part representations towards or into the deprived S1 hand cortex may have taken

place. Whether reorganisation and preservation of the original function can simultaneously take place within the same cortical area therefore remains a topic for further investigation. It is possible that reorganisation and preservation of the original function could co-occur within cortical areas. Indeed, non-human primate studies demonstrated that remapping observed in S1 actually reflects reorganisation in subcortical areas of the somatosensory pathway, principally the brainstem (*Chand and Jain, 2015*; *Kambi et al., 2014*). As such, the deprived S1 area receives reorganised somatosensory inputs upon tactile stimulation of neighbouring intact body parts. This would simultaneously allow the original S1 representation of the deprived body part to be preserved, as observed in our results when we directly probed the deprived S1 hand area through attempted finger movements.

Together, our findings indicate that in the first years after a tetraplegia the somatotopic S1 hand representation is preserved and can be activated through attempted movements even in the absence of retained sensory hand function, motor hand function, and spared spinal tissue bridges. These preserved S1 finger maps could be exploited in a functionally meaningful manner by rehabilitation approaches that aim to establish new functional connections between the brain and the hand after a tetraplegia, for example, through neuroprosthetic limbs or advanced exoskeletons that are directly controlled by the brain (*Ajiboye et al., 2017*; *Armenta Salas et al., 2018*; *Bouton et al., 2016*; *Lebedev and Nicolelis, 2017*).

**Table 1.** Demographic and clinical details.

Tetraplegic patients are ordered according to their retained upper-limb sensory and motor function (assessed using the Graded Redefined Assessment of Strength, Sensibility and Prehension test [GRASSP]). Sex: F, female; M, male; Age, age in years; AIS grade, American Spinal Injury Association (ASIA) Impairment Scale grade defined based on the International Standards for Neurological Classification of Spinal Cord Injury (ISNCSCI); A, complete; B, sensory incomplete; C, motor incomplete; D, motor incomplete; E, normal; Neurological level of injury, defined based on the ISNCSCI; dominant hand, defined using the Edinburgh handedness inventory: L, left; R, right; GRASSP, Graded Redefined Assessment of Strength, Sensibility and Prehension (maximum score: 232 points); tested side, side with the lowest score on the GRASSP measurement; GRASSP motor/sensory score of the tested upper limb (maximum scores: 50/24; see *Table 2* for further details).

|  | Sex | Age | Years since injury | AIS grade | Cause of injury | Neurological level of injury | Dominant hand | GRASSP score | Hand tested | GRASSP tested side motor/sensory |
|---|---|---|---|---|---|---|---|---|---|---|
| S01 | M | 32 | 4 | A | Trauma | C4 | L | 21 | L | 9/0 |
| S02 | M | 52 | 32 | A | Trauma | C5 | R | 78 | L | 16/5 |
| S03 | M | 35 | 4 | A | Trauma | C4 | R | 90 | L | 16/17 |
| S04 | M | 41 | 19 | A | Trauma | C6 | L | 105 | L | 19/11 |
| S05 | M | 52 | 10 | A | Trauma | C2 | L | 118 | R | 23/0 |
| S06 | M | 67 | 26 | A | Trauma | C4 | R | 119 | L | 29/2 |
| S07 | M | 57 | 33 | C | Trauma | C5 | L | 145 | R | 25/19 |
| S08 | F | 67 | 4 | D | Trauma | C5 | L | 173 | L | 41/9 |
| S09 | M | 59 | 12 | D | Trauma | C2 | R | 187 | R | 43/9 |
| S10 | M | 42 | 2 | D | Trauma | C4 | R | 187 | R | 41/13 |
| S11 | M | 58 | 0.5 | D | Ischaemic | C4 | R | 194 | R | 37/17 |
| S12 | M | 71 | 16 | D | Trauma | C7 | R | 196 | R | 32/24 |
| S13 | M | 65 | 1 | D | Trauma | C2 | R | 218 | R | 42/24 |
| S14 | M | 74 | 6 | D | Surgery | C4 | L | 220 | R | 47/24 |

# Materials and methods

## Participants

15 chronic (i.e. > 6 months post injury) tetraplegic patients were recruited and 14 patients completed the measurements (mean age ± s.e.m. = 55 ± 3.6 years ; one female; six dominant left-handers; see *Table 1* for demographic and clinical details). Patient inclusion criteria were as follows: aged 18–75 years, no MRI contraindications, at least 6 months post SCI, no neurological impairment or body function impairments not induced by SCI, and able to provide informed consent. 18 age-, sex-, and handedness-matched able-bodied control participants (age = 56 ± 3.6 years; one female; five dominant left-handers) also participated in this study. Control participant inclusion criteria were as follows: aged 18–75 years, no MRI contraindications, no impairment of body function induced by SCI, no neurological illness, no hand impairments, and able to provide informed consent.

Participants' informed consent was obtained according to the Declaration of Helsinki prior to study onset. Ethical approval was granted by the Kantonale Ethikkommission Zürich (KEK-2018-00937). This study is registered on clinicaltrials.gov under NCT03772548. Two patients and one control participant were scanned twice due to excessive head motion during fMRI acquisition or suboptimal slice placement. One patient withdrew from the study prior to study completion. Data of one control participant were distorted and not usable for analysis. All data relating to these participants were discarded from all data analysis. Patient S02 did not complete the travelling wave measurements due to time constraints.

## Clinical characterisation

Behavioural testing was conducted in a separate session. We used the International Standards for Neurological Classification of Spinal Cord Injury (ISNCSCI) to neurologically classify patients' completeness of injury and impairment level. We used the GRASSP assessment to define sensory and motor integrity of the upper limbs (*Kalsi-Ryan et al., 2012*). Each upper limb's maximum score is 116 and refers to healthy conditions. We determined each patient's most impaired upper limb according to the GRASSP. Note that GRASSP motor scores reflect overall upper-limb motor function (i.e. including arm and shoulder functioning; see *Table 2* for muscle-specific GRASSP scores). GRASSP sensory scores are hand specific.

## fMRI tasks

We employed two separate paradigms to uncover fine-grained somatotopic hand representations using fMRI: first, we used a travelling wave paradigm to investigate the somatotopic hand layout on the S1 cortical surface (*Besle et al., 2013*; *Kolasinski et al., 2016a*). Second, we employed a blocked design and RSA that takes into account the entire fine-grained activity pattern of each finger (i.e. including the representational inter-finger relationships; *Ejaz et al., 2015*; *Kriegeskorte et al., 2008*).

Participants were visually cued to perform individual finger movements while their palm was positioned up. Patients were instructed to perform the fMRI tasks with their most impaired upper limb (identified using the GRASSP, see previous section). Controls' tested hands were matched to the patients. Due to their injury, not all patients were able to make overt finger movements. In these cases, patients were carefully instructed by the experimenter to make attempted (i.e. not imagined) finger movements. The experimenter explained that although it is not possible for the patient to perform an overt movement, attempting to perform the movement may still produce a motor command in the brain. Importantly, complete paraplegic patients are able to distinguish between attempted and imagined movements with their paralysed body part (*Cramer et al., 2005*; *Hotz-Boendermaker et al., 2008*; *Sabbah et al., 2002*). Furthermore, attempted foot movements activated SCI patients' motor network similarly to controls performing overt foot movements (*Cramer et al., 2005*; *Hotz-Boendermaker et al., 2008*; *Sabbah et al., 2002*).

Participants saw five horizontally aligned white circles, corresponding to the five fingers, via a visual display viewed through a mirror mounted on the head coil. For participants moving their left hand, the leftmost and rightmost circles corresponded to the thumb and little finger, respectively. For participants moving their right hand, the leftmost circle corresponded to the little finger and the rightmost circle to the thumb. To cue a finger movement, the circle corresponding to this finger turned red. Participants performed self-paced flexion/extension movements with the cued finger for the duration

**Table 2. GRASSP motor sub-scores.**

Each muscle was tested with resistance through its full range of motion and given a muscle grade between 0 and 5: 0, flaccid motion; 1, flicker motion; 2, full range of motion with gravity eliminated; 3, full range of motion against gravity; 4, full range of motion with moderate resistance; 5, full range of motion with maximal resistance (*Kalsi-Ryan et al., 2012*).

| | Elbow flexion | Shoulder (deltoideus) | Wrist extension | Elbow extension | Fingers 2–5 extension | Thumb opposition | Thumb flexion | Middle finger flexion | Little finger abduction | Index finger abduction |
|---|---|---|---|---|---|---|---|---|---|---|
| S01 | 5 | 4 | 0 | 0 | 0 | 0 | 0 | 0 | 0 | 0 |
| S02 | 5 | 5 | 4 | 2 | 0 | 0 | 0 | 0 | 0 | 0 |
| S03 | 4 | 4 | 4 | 3 | 1 | 0 | 0 | 0 | 0 | 0 |
| S04 | 5 | 5 | 5 | 4 | 0 | 0 | 0 | 0 | 0 | 0 |
| S05 | 4 | 1 | 3 | 4 | 3 | 3 | 1 | 1 | 1 | 2 |
| S06 | 5 | 5 | 5 | 5 | 1 | 1 | 4 | 3 | 0 | 0 |
| S07 | 5 | 5 | 5 | 5 | 1 | 1 | 1 | 0 | 1 | 1 |
| S08 | 5 | 4 | 5 | 4 | 4 | 5 | 5 | 4 | 4 | 1 |
| S09 | 5 | 4 | 4 | 4 | 4 | 4 | 5 | 5 | 4 | 4 |
| S10 | 5 | 5 | 4 | 4 | 3 | 4 | 4 | 4 | 4 | 4 |
| S11 | 5 | 5 | 4 | 2 | 5 | 4 | 5 | 5 | 1 | 1 |
| S12 | 5 | 5 | 5 | 5 | 4 | 1 | 4 | 1 | 1 | 1 |
| S13 | 5 | 4 | 4 | 4 | 5 | 4 | 5 | 3 | 4 | 4 |
| S14 | 5 | 5 | 5 | 5 | 5 | 5 | 5 | 4 | 4 | 4 |

of the colour change. Instructions were delivered using Psychtoolbox (v3) implemented in MATLAB (v2014). Head motion was minimised using over-ear MRI-safe headphones or padded cushions.

The travelling wave paradigm involved individuated finger movements in a set sequence. Each 10 s finger movement block was immediately followed by a movement block of a neighbouring finger. The forward sequence cycled through the fingers: thumb-index-middle-ring-little. To account for order-related biases due to the set movement cycle and sluggish haemodynamic response, we also collected data using a backward sequence: the backward sequence cycled through the movements in a reverse of the forward sequence: little-ring-middle-index-thumb fingers. The forward and backward sequences were employed in separate runs. A run lasted 6 min and 4 s, during which a sequence was repeated seven times. The forward and backward runs were repeated twice, with a total duration of 24 min and 16 s.

The blocked design consisted of six conditions: movement conditions for each of the five fingers and a rest condition. Finger movement instructions were as described above, and the word 'Rest' indicated the rest condition. A movement block lasted 8 s, and each condition was repeated five times per run in a counterbalanced order. Each run comprised a different block order and had a duration of 4 min and 14 s. We acquired four runs, with a total duration of 16 min and 56 s.

## MRI acquisition

MRI data were acquired using a Philips 3 Tesla Ingenia system (Best, The Netherlands) with a 17-channel HeadNeckSpine or, in case of participant discomfort due to the coil's narrowness, a 15-channel Head-Spine coil. Anatomical T1-weighted images covering the brain and cervical spinal cord were acquired using the following acquisition parameters: 0.8 mm$^3$ resolution, repetition time (TR) = 9.3 ms, echo time (TE) = 4.4 ms, and flip angle 8°. Anatomical T2-weighted images of the cervical spinal cord were acquired sagittally using the following acquisition parameters: 1 × 1 × 3 mm resolution, TR = 4500 ms, TE = 85 ms, flip angle = 90°, and slice gap = 0.3 mm, 15 slices. Task-fMRI data were acquired using an echo-planar-imaging (EPI) sequence with partial brain coverage: 22 sagittal slices were centred on the anatomical location of the hand knob with coverage over the thalamus and brainstem. We used the following acquisition parameters: 2 mm$^3$ resolution, TR = 2000 ms, TE = 30 ms, flip angle = 82°, and SENSE factor = 2.2. We acquired 182 and 127 volumes for each of the travelling wave and blocked design runs, respectively.

## fMRI analysis

fMRI analysis was implemented using FSL v6.0 (https://fsl.fmrib.ox.ac.uk/fsl/fslwiki), Advanced Normalization Tools (ANTs) v2.3.1 (http://stnava.github.io/ANTs), the RSA toolbox (*Nili et al., 2014*; *Wesselink and Maimon-Mor, 2017*), and MATLAB (R2018a). Cortical surface visualisations were realised using FreeSurfer (https://surfer.nmr.mgh.harvard.edu/; *Dale et al., 1999*; *Fischl et al., 2001*) and Connectome Workbench (https://www.humanconnectome.org/software/connectome-workbench).

## fMRI preprocessing

Common preprocessing steps were applied using FSL's Expert Analysis Tool (FEAT). The following preprocessing steps were included: motion correction using MCFLIRT (*Jenkinson et al., 2002*), brain extraction using automated brain extraction tool BET (*Smith, 2002*), spatial smoothing using a 2 mm full-width-at-half-maximum (FWHM) Gaussian kernel, and high-pass temporal filtering with a 100 s (blocked design runs) or 90 s (travelling wave runs) cut-off.

## Image registration

Image co-registration was done in separate, visually inspected, steps. For each participant, a midspace was calculated between the four blocked design runs, that is, an average space in which images are minimally reoriented. We then transformed all fMRI data to this midspace using purely rigid probability mapping in ANTs. Next, we registered each participant's midspace to the T1-weighted image, initially using 6 degrees of freedom and the mutual information cost function, and then optimised using boundary-based registration (BBR; *Greve and Fischl, 2009*) Each co-registration step was visually inspected and, if needed, manually optimised using blink comparison in Freeview.

## Travelling wave analysis

The travelling wave approach is characterised by set finger movement cycles that are expected to result in neighbouring cortical activations. It is designed to capture voxels that show preferential activity to

one condition, above and beyond all other conditions (i.e. winner-takes-all principle; testing for finger selectivity). The travelling wave approach is especially powerful to reveal the smooth progression of neighbouring representations that are specific for topographic maps. This technique is therefore frequently used to uncover retinotopic (*DeYoe et al., 1996*; *Engel et al., 1997*; *Sereno et al., 1995*), somatotopic (*Besle et al., 2013*; *Kikkert et al., 2016*; *Kolasinski et al., 2016a*; *Mancini et al., 2012*; *Zeharia et al., 2015*), and tonotopic representations (*Da Costa et al., 2015*; *Talavage et al., 2004*). Importantly, S1 finger movement somatotopy assessed using the travelling wave approach is highly consistent across scanning sessions (*Kolasinski et al., 2016a*).

Travelling wave analysis was conducted separately for each participant, closely following procedures previously described in *Kikkert et al., 2016*. A reference model was created using a gamma haemodynamic response function (HRF) convolved boxcar, using a 10 s 'on' (a single finger movement duration) and 40 s 'off' period (movement duration of all other fingers). This reference model was then shifted in time to model activity throughout the full movement cycle. Since we had a 2 s TR and a 50 s movement cycle, the reference model was shifted 25 times.

Within each individual run, each voxel's preprocessed BOLD signal time course was cross-correlated with each of the 25 reference models. This resulted in 25 r-values per voxel per run that were normalised using a Fisher r-to-z transformation. To create finger-specific (i.e. hard-edged) maps, we assigned individual lags to specific fingers and averaged the z-values across these lags. This resulted in five z-values, one for each finger per voxel per run. These z-values were averaged across runs per voxel and per finger assignment. As a result, each individual voxel now contained five averaged z-values, one for each finger. Next, a winner-take-all approach was used to assign each voxel to one finger exclusively based on the maximum z-value, providing us with finger specificity.

To visualise the smooth gradient of progression across fingers, we produced lag-specific maps. The backward runs' standardised cross-correlation z-values were lag-reversed and averaged with the forward runs per voxel and per lag. This resulted in 25 z-values per voxel (one per lag). Next, we used a winner-take-all principle to find the maximum z-value across lags for each voxel, providing us with lag specificity.

Cortical surface projections were constructed from participant's T1-weighted images. The winner-take-all finger-specific and lag-specific gradient maps were projected onto the cortical surface using cortical-ribbon mapping. Thresholding was applied to the winner-take-all finger-specific maps and lag-specific maps on the cortical surface using a false discovery criterion q < 0.05 based on the native (3D) values. The false discovery rate (FDR) thresholded finger-specific maps were combined to form a hand map. Within this hand map, the lag-specific map was used to visualise the smooth gradient of progression across fingers. We were unable to find a characteristic hand map in one patient who anecdotally reported post hoc that he was unable to performed attempted finger movements.

## Inter-participant probability of finger selectivity

To visualise inter-participant consistency of somatotopic finger-selective representations, we calculated cortical activation probability maps. To ensure that the tested hemisphere was consistently aligned for all participants, we first flipped the acquired T1-weighted images and the travelling wave winner-take-all finger-specific maps along the x-axis for the left-hand tested participants. For these left-hand tested participants, we created new cortical surface projections using their flipped T1-weighted images. Each participant's cortical surface was then inflated into a sphere and aligned to the FreeSurfer 2D average atlas using sulcal depth and curvature information. The travelling wave winner-take-all finger-specific maps were resampled to the FreeSurfer 2D average atlas and thresholded using a false discovery criterion q < 0.05 based on the native (3D) values. We then calculated finger-specific probability maps for the control and SCI patient groups, separately.

## Spatial correspondence of finger maps over time: DOC analysis

To confirm that the travelling wave finger-specific maps did not represent random noise, we quantified spatial consistency of finger preference between two halves of the data using the Dice overlap coefficient (DOC; *Dice, 1945*). The DOC calculates the spatial overlap between two representations relative to the total area of these representations. The DOC ranges from 0 (no spatial overlap) to 1 (perfect spatial overlap). If A and B represent the areas of two representations, then the DOC is expressed as

$$\frac{2x\ |A \cap B|}{|A| + |B|}$$

We followed previously described procedures for calculating the DOC between two halves of the travelling wave data (**Kikkert et al., 2016**; **Kolasinski et al., 2016a**; **Sanders et al., 2019**). The averaged finger-specific maps of the first forward and backward runs formed the first data half. The averaged finger-specific maps of the second forward and backward runs formed the second data half. The finger-specific clusters were minimally thresholded ($Z > 2$) on the cortical surface and masked using an S1 ROI that was created based on Brodmann area parcellation using FreeSurfer. We used minimally thresholded finger-specific clusters for DOC analysis to ensure that we were sensitive to overlaps that would be missed when using high thresholds (see **Figure 2—figure supplement 1** for a visualisation of the minimally thresholded split-half hard-edged finger maps used to calculate the DOC). Note that the same results were found when thresholding the finger-specific clusters using an FDR q < 0.05 criterion (see **Figure 2—figure supplement 2**). The DOC was calculated between same, neighbouring, and non-neighbouring fingers across the two data halves (see **Figure 2C**).

If the finger maps would be spatially consistent and represent true finger selectivity, then one would expect a higher DOC between pairs of 'same' fingers across the two data halves compared to neighbouring and non-neighbouring finger pairs. One would further expect to find a somatotopic relationship in the DOCs: that is, a higher DOC between neighbouring compared to non-neighbouring finger pairs. We tested whether the somatotopic relationship in the DOCs was different in controls and patients using a robust mixed ANOVA with a within-participants factor for finger pair (three levels: same, neighbouring and non-neighbouring finger pairs) and a between-participants factor for group (two levels: controls and SCI patients).

## Univariate analysis

To assess univariate task-related activity of the blocked design data, time-series statistical analysis was carried out per run using FMRIB's Improved Linear Model (FILM) with local autocorrelation correction, as implemented in FEAT. We obtained activity estimates using a general linear modelling (GLM) based on the double-gamma HRF and its temporal derivative. Each finger movement condition was contrasted with rest. A further contrast was defined for overall task-related activity by contrasting all movement conditions with rest. A fixed effects higher-level analysis was ran for each participant to average across runs.

We defined an S1 hand ROI by converting the S1 ROI used to calculate split-half consistency to volumetric space. Any holes were filled and non-zero voxels were mean dilated. Next, the axial slices spanning 2 cm medial/lateral to the hand knob (**Yousry et al., 1997**) were identified on the 2 mm MNI standard brain (min-max MNI z-coordinates = 40–62). This mask was non-linearly transformed to each participant's native structural space. Finally, we used this mask to restrict the S1 ROI and extracted an S1 hand area ROI. The percent signal change for overall task-related activity was then extracted for voxels underlying this S1 hand ROI per participant. A similar analysis was used to investigate overall task-related activity in an M1 hand ROI (see **Figure 3—figure supplement 1**). We further compared activity levels in finger-specific ROIs in S1 between groups and conducted a geodesic distance analysis to assess whether the finger representations of SCI patients were aligned differently and/or shifted compared to the control group (see Appendix 1).

## Representational similarity analysis

While the traditional travelling wave approach is powerful to uncover the somatotopic finger arrangement, a fuller description of hand representation can be obtained by taking into account the entire fine-grained activity pattern of all fingers. RSA-based inter-finger overlap patterns have been shown to depict the invariant representational structure of fingers better than the size, shape, and exact location of the areas activated by finger movements (**Ejaz et al., 2015**). RSA-based measures are furthermore not prone to some of the problems of measurements of finger selectivity (e.g. dependence on map thresholds). We estimated inter-finger overlap using RSA. Note that it is also possible to estimate somatotopic overlap from travelling wave data using an iterated Multigrid Priors (iMGP) method and population-receptive field modelling (**Da Rocha Amaral et al., 2020**; **Puckett et al., 2020**). We computed the distance between the activity patterns measured for each finger pair within

the S1 hand ROI using the cross-validated squared Mahalanobis distance (or crossnobis distance; *Nili et al., 2014*). We extracted the blocked design voxel-wise parameter estimates (betas) for each finger movement condition versus rest (identified in the univariate analysis) and the model fit residuals under the S1 hand ROI. We prewhitened the extracted betas using the model fit residuals. We then calculated the cross-validated squared Mahalanobis distances between each possible finger pair, using our four runs as independent cross-validation folds, and averaged the resulting distances across the folds. If it is impossible to statistically differentiate between conditions (i.e. when this parameter is not represented in the ROI), the expected value of the distance estimate would be 0. If it is possible to distinguish between activity patterns, this value will be larger than 0.

The distance values for all finger pairs were assembled in a representational dissimilarity matrix (RDM), with a width and height corresponding to the five finger movement conditions. Since the RDM is mirrored across the diagonal with meaningless zeros on the diagonal, all statistical analyses were conducted on the 10 unique off-diagonal values of the RDM. We first estimated the strength of the finger representation or 'finger separability' by averaging the 10 unique off-diagonal values of the RDM. If there is no information in the ROI that can statistically distinguish between the finger conditions, then due to cross-validation the expected separability would be 0. If there is differentiation between the finger conditions, the separability would be larger than 0 (*Nili et al., 2014*). Note that this does not directly indicate that this region contains topographic information, but rather that this ROI contains information that can distinguish between the finger conditions. To further ensure that our S1 hand ROI was activated distinctly for different fingers, we created a CSF ROI that would not contain finger-specific information. We repeated our RSA analysis in this ROI and statistically compared the separability of the CSF and S1 hand area ROIs. Second, we tested whether the inter-finger distances were different between controls and patients using a robust mixed ANOVA with a within-participants factor for finger pair (10 levels) and a between-participants factor for group (two levels: controls and SCI patients). Third, we estimated the somatotopic typicality (or normality) of each participant's RDM by calculating a Spearman correlation with a canonical RDM. We followed previously described procedures for calculating the typicality score (*Dempsey-Jones et al., 2019*; *Ejaz et al., 2015*; *Kieliba et al., 2021*; *Wesselink et al., 2019*). The canonical RDM was based on 7T finger movement fMRI data in an independently acquired cohort of healthy controls (n = 8). The S1 hand ROI used to calculate this canonical RDM was defined similarly as in the current study (see *Wesselink et al., 2021* for details). Note that results were unchanged when calculating typicality scores using a canonical RDM based on the averaged RDM of the age-, sex-, and handedness-matched control group tested in this study (see Appendix 1). The typicality scores were Fisher r-to-z transformed prior to statistical analysis (the $r_s$ typicality scores are used solely for visualisation). Controls' and SCI patients' typicality scores were compared to each other and to those of a group of individuals with congenital hand loss (n = 13), hereafter one-handers, obtained in another study (data publicly available on https://osf.io/gmvua/; *Wesselink et al., 2019*). Congenital one-handers are born without a hand and do not have an S1 hand representation contralateral to the missing hand.

Finally, we performed multidimensional scaling (MDS) to visualise the distance structure of the RDM in an intuitive manner. MDS projects the higher-dimensional RDM into a lower-dimensional space, while preserving the inter-finger distance values as well as possible (*Borg and Groenen, 2005*). MDS was performed for each individual participant and then averaged per group after Procrustes alignment to remove arbitrary rotation induced by MDS.

## Structural MRI analysis
### Midsagittal tissue bridges analysis
We used sagittal T2w structural images of the cervical spinal cord at the lesion level to quantify spared tissue bridges. It has been shown that already in the sub-acute stage after SCI oedema and haemorrhage have largely resolved and hyperintense signal changes reliably reflect intramedullary neural damage (*Huber et al., 2017*; *Pfyffer et al., 2019*; *Vallotton et al., 2019*). We closely followed previously described procedures (*Huber et al., 2017*; *Pfyffer et al., 2019*; *Vallotton et al., 2019*). We used Jim 7.0 software (Xinapse Systems, Aldwincle, UK) for manual lesion segmentation at the lesion level, for which high intra- and interobserver reliability has previously been reported (*Huber et al., 2017*; *Pfyffer et al., 2019*). The experimenter conducting the manual segmentation was blinded to patient identity. We only included patients' T2w scans if the lesion (i.e. hyperintense, CSF-filled cystic cavity)

was clearly visible on the midsagittal slice. We excluded images of two patients with metal artefacts or insufficient data quality which would not allow a reliable quantification of lesion measures. Tissue bridges were defined as the relatively hypointense intramedullary region between the hyperintense CSF on one side and the cystic cavity on the other side. We assessed the width of ventral and dorsal tissue bridges on the midsagittal slice and summed these to get the total width of tissue bridges.

## Cervical cross-sectional spinal cord area analysis

We used Jim 7.0 software (Xinapse Systems) to extract the cross-sectional spinal cord area at cervical level C2/3 of the spinal cord from the sagittal T2w scans. We used multi-planar reconstruction of sagittal images, resulting in 10 contiguous axial slices at C2/3 with a thickness of 2 mm (*Losseff et al., 1996*). Using the active-surface model from *Horsfield et al., 2010*, the cross-sectional spinal cord area was calculated semi-automatically for every slice and averaged over all 10 slices.

## Statistical analysis

Statistical analysis was carried out using SPSS (v25). Standard approaches were used for statistical analysis, as mentioned in the Results section. If normality was violated (assessed using the Shapiro–Wilk test), non-parametric statistical testing or robust ANOVAs (in RStudio v1.4; WRS2 package; *Mair and Wilcox, 2020*) were used. We used a Crawford–Howell t-test to compare single patients to the congenital and control groups (*Corballis, 2009*). All testing was two-tailed, and corrected p-values were calculated using the Benjamini–Hochberg procedure to control the FDR with q < 0.05. The correlational analysis was considered exploratory and we did not correct for multiple comparisons in this analysis.

Bayesian analysis was carried out using JASP (v0.12.2) for the main comparisons to investigate support for the null hypothesis with the Cauchy prior width set at 0.707 (i.e. JASP's default). Following the conventional cut-offs, a BF smaller than 1/3 is considered substantial evidence in favour of the null hypothesis. A BF greater than 3 is considered substantial evidence, and a BF greater than 10 is considered strong evidence in favour of the alternative hypothesis. A BF between 1/3 and 3 is considered weak or anecdotal evidence (*Dienes, 2014*; *Kass and Raftery, 1995*).

## Acknowledgements

We thank our participants for taking part in the study. We thank Lydia Kämpf, Nicolin Gauler, and Silvia Hofer for their assistance with data collection, Daan Wesselink for help with data analysis, and Tamar Makin for comments on the manuscript.

## Additional information

### Funding

| Funder | Grant reference number | Author |
| --- | --- | --- |
| Swiss National Science Foundation | 320030_175616 | Nicole Wenderoth |
| ETH Zurich Postdoctoral Fellowship Program | | Sanne Kikkert |
| Swiss National Science Foundation | PCEFP3_181362 / 1 | Patrick Freund |

The funders had no role in study design, data collection and interpretation, or the decision to submit the work for publication.

### Author contributions

Sanne Kikkert, Conceptualization, Formal analysis, Funding acquisition, Investigation, Project administration, Software, Supervision, Visualization, Writing - original draft, Writing – review and editing; Dario Pfyffer, Conceptualization, Formal analysis, Investigation, Software, Writing – review and editing; Michaela Verling, Investigation, Project administration, Writing – review and editing; Patrick Freund,

Conceptualization, Supervision, Writing – review and editing; Nicole Wenderoth, Conceptualization, Funding acquisition, Resources, Supervision, Writing – review and editing

### Author ORCIDs
Sanne Kikkert  http://orcid.org/0000-0001-9952-4864
Dario Pfyffer  http://orcid.org/0000-0002-2406-9251
Patrick Freund  http://orcid.org/0000-0002-4851-2246
Nicole Wenderoth  http://orcid.org/0000-0002-3246-9386

### Ethics
Participants' informed consent was obtained according to the Declaration of Helsinki prior to study onset. Ethical approval was granted by the Kantonale Ethikkommission Zürich (KEK-2018-00937).

### Decision letter and Author response
Decision letter https://doi.org/10.7554/eLife.67713.sa1
Author response https://doi.org/10.7554/eLife.67713.sa2

## Additional files

### Supplementary files
• Transparent reporting form
• Reporting standard 1. STROBE checklist.

### Data availability
Full details of the experimental protocol are available on clinicaltrials.gov under the number NCT03772548. Data is shared on https://osf.io/e8u95/.

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

# Appendix 1

## Percent signal change in finger-specific clusters

To assess whether finger movement activity levels were different between patients and controls, we created finger-specific ROIs and extracted the activity level of the corresponding finger movement for each participant. To create the finger-specific ROIs, we thresholded the probability finger surface maps that were created from the travelling wave data of the control group (see main text) at 25% (i.e. meaning that at least 5 out of 18 control participants needed to significantly activate a vertex for this vertex to be included in the ROI) and binarised. We then used the separately acquired blocked design data to extract the finger movement activity levels underlying these finger-specific ROIs. We first flipped the contrast images resulting from the fixed effects analysis (i.e. that was ran to average across the four blocked design runs) along the x-axis for the left-hand tested participants. Each participant's contrast maps were then resampled to the FreeSurfer 2D average atlas, and the averaged z-standardised activity level was extracted for each finger movement vs. rest contrast underlying the finger-specific ROIs.

### A) 25% probability finger selectivity maps of the control group

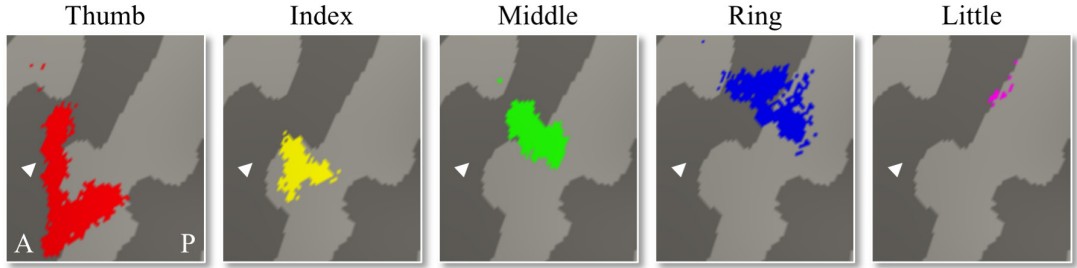

### B) Finger movement activity in the corresponding finger-specific ROIs

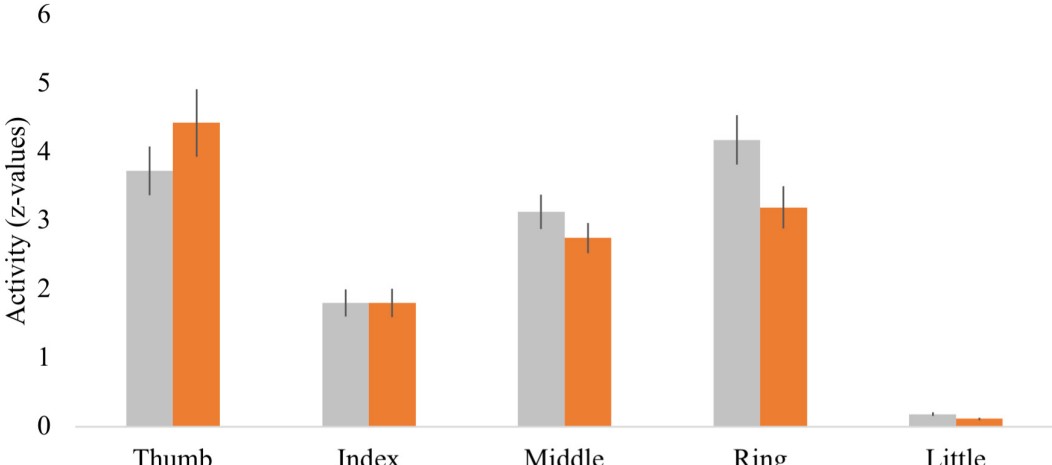

**Appendix 1—figure 1.** Finger-specific activity levels in finger-specific regions of interest (ROIs). (**A**) Finger-specific ROIs were based on the control group's binarised 25% probability travelling wave finger selectivity maps. White arrows indicate the central sulcus. A: anterior; P: posterior. (**B**) Finger movement activity levels in the corresponding finger-specific ROIs. There were no significant differences in activity levels between thetetraplegic patient and control groups. Controls are projected in grey; patients are projected in orange. Error bars show the standard error of the mean.

We compared the activity levels for each finger movement in the corresponding finger ROI (i.e. thumb movement activity in the thumb ROI, index finger movement activity in the index finger ROI, etc.) between groups. After correction for multiple comparisons, there was no significant difference between groups for the thumb (U = 93, p=0.37), index ($t_{(30)}$ = −0.003, p=0.99), middle ($t_{(30)}$ = 1.11, p=0.35), ring ($t_{(30)}$ = 2.02, p=0.13), or little finger ($t_{(30)}$ = 2.14, p=0.20).

## Geodesic distance analysis

To assess whether the finger representations of the SCI patients were aligned differently and/or shifted compared to the control participants, the cortical distance was calculated between each participant's peak finger activation and a reference anchor in the S1 foot cortex. We used the blocked design data for this analysis as the travelling wave paradigm does not lend itself to accurately extract peak finger activation locations due to the consecutive movement task.

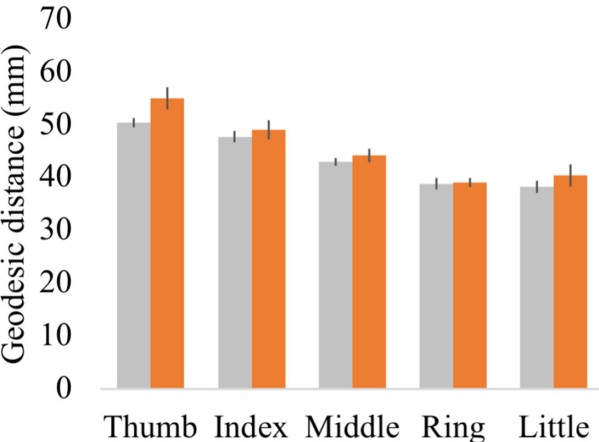

**Appendix 1—figure 2.** Geodesic distance between each finger's peak activated vertex and a cortical anchor. Cortical geodesic distances were calculated between a reference anchor in the S1 foot cortex (MNI coordinates: −16.93, −32.06, 73.80) and the peak activated vertex per finger movement for each participant. Controls are projected in grey; spinal cord injury (SCI) patients are projected in orange. Error bars show the standard error of the mean.

Each participant's fixed effect contrast maps that were already resampled to the FreeSurfer 2D average atlas were masked using the S1 hand ROI. The geodesic distance was then calculated between the peak activated vertex in the S1 hand cortex and a reference anchor in the S1 foot cortex (MNI coordinates: −16.93, −32.06, 73.80). Note that geodesic distance measures calculated onto average cortical surfaces are not confounded by inter-subject variability in gyrification, follow the anatomical constraints of the brain (i.e. do not cross white matter boundaries), and allow comparison between different subjects. This is a key advantage of this approach over using Euclidean distance measures (*Mancini et al., 2019*). A robust mixed ANOVA with a within-participants factor for finger (five levels: thumb, index, middle, ring, and little finger) and a between-participants factor for group (two levels: controls and patients) showed that, as expected, the geodesic distances were significantly different across fingers (i.e. a main effect for fingers; $F_{(1,18.81)} = 4.29$, $p=0.05$). While the peak finger vertexes were located more lateral for the patient compared to the control group (i.e. a main effect for group; $F_{(4,13.93)} = 85.66$, $p<0.001$), the geodesic distance pattern across the fingers was not significantly different across groups (i.e. no significant finger by group interaction; $F_{(4,13.93)} = 0.91$, $p=0.48$). When testing for group differences per finger, the BF only revealed inconclusive evidence (BF >0.34 and < 1.88; note that we could not run a Bayesian ANOVA due to normality violations).

## Typicality analysis using a canonical RDM based on the controls' average RDM

To ensure that our typicality results did not change when using a canonical inter-finger RDM based on the age-, sex-, and handedness-matched subjects tested in this study, we recalculated the typicality scores of all participants using the averaged inter-finger RDM of our control sample as the canonical RDM. We found a strong and highly significant correlation between the typicality scores calculated using the canonical inter-finger RDM from the independent dataset (reported in the main text) and the typicality scores calculated using our controls' average RDM. This was true for both the SCI patient (rs = 0.92, p<0.001) and control groups (rs = 0.78, p<0.001).

We then repeated all typicality analysis reported in the main text. As expected, we found the same results using the typicality scores calculated using our controls' average RDM as when using the canonical inter-finger RDM from the independent dataset: There was a significant difference in typicality between tetraplegic patients, healthy controls, and congenital one-handers ($H_{(2)}$ = 27.61, p<0.001). We further found significantly higher typicality in controls compared to congenital one-handers (U = 0, p<0.001; $BF_{10}$ = 76.11). Importantly, the typicality scores of the SCI patients were significantly higher than the congenital one-handers (U = 2, p<0.001; $BF_{10}$ = 50.98), but not significantly different from the controls (U = 94, p=0.24; $BF_{10}$ = 0.55). Number of years since SCI significantly correlated with hand representation typicality ($r_s$ = −0.54, p=0.05) and patients with more retained GRASSP motor function of the tested upper limb had more typical hand representations in S1 ($r_s$ = 0.58, p=0.03). There was no significant correlation between S1 hand representation typicality and GRASSP sensory function of the tested upper limb, spared midsagittal spinal tissue bridges at the lesion level, or cross-sectional spinal cord area ($r_s$ = 0.40, p=0.15, $r_s$ = 0.50, p=0.10, and $r_s$ = 0.48, p=0.08, respectively). An exploratory stepwise linear regression analysis revealed that years since SCI significantly predicted hand representation typicality in S1 with $R^2$ = 0.33 ($F_{(1,10)}$ = 4.98, p=0.05). Motor function, sensory function, spared midsagittal spinal tissue bridges at the lesion level, and spinal cord area did not significantly add to the prediction (t = 1.31, p=0.22, t = 1.62, p=0.14, t = 1.70, p=0.12, and t = 1.09, p=0.30, respectively).

