## [Decision Letter]

**Acceptance summary:**

This paper demonstrates that the spatial organization of the somatosensory cortex deteriorates very slowly following spinal cord injury that results in tetraplegia. The findings contribute to a developing story on how sensory representations are formed, changed and maintained and has implications for the development of brain-machine interfaces.

**Decision letter after peer review:**

Thank you for submitting your article "Finger somatotopy is preserved after tetraplegia but deteriorates over time" for consideration by *eLife*. Your article has been reviewed by 3 peer reviewers, and the evaluation has been overseen by Andrew Pruszynski as Reviewing Editor and Richard Ivry as the Senior Editor. The following individuals involved in review of your submission have agreed to reveal their identity: Sliman J Bensmaia (Reviewer #3).

The reviewers have discussed their reviews with one another, and the Reviewing Editor has drafted this letter to help you prepare a revised submission.

Essential revisions:

1) The authors need to provide a clearer hypothesis about the mechanistic changes of the sensorimotor maps in the patients. This requires a thorough description of prior results and how these prior results motivated the present study and the analyses conducted. The reviewers emphasize that this will also require additional analyses that bring the present study into alignment with previous work. In this respect, Reviewer 1 and 2 indicate specific figures and analyses that need to be done and presented in a revised manuscript.

2) There is substantial concern that the key "typicality" analysis is not valid. As detailed by Reviewer 2 there are previous studies that have approached this issue but the present manuscript largely ignores this previous literature and uses their own ad hoc approach. The issue of defining typicality needs to be meaningfully (analytically) addressed in reference to previous established approaches.

3) There are several analytical points that require attention, either because the choices are unclear or that critical details are not reported.

4) Provide a more thorough Discussion, especially in the context of alternative explanations for the present findings. The reviewers all raise important concerns that the present account is very motor-effector dominated and dismisses other potential explanations. In particular, pay close attention to the comments raised by Reviewer #1 in terms of attentional mechanisms and how they may confound the present empirical results. This limitation needs to be addressed either experimentally, or given that possibility is unlikely, explicitly in the Discussion.

*Reviewer #1 (Recommendations for the authors):*

Abstract: It is unclear what evidence serves the foundation for the authors' claim that rehabilitation approaches would be most promising when applied within the first years after injury. I can see why this might be speculated, but is there any previous work showing that those with in-tact hand representations are more conducive to rehabilitation than those with deteriorated representations?

Lines 106-116: This section somewhat gives the impression that complete digit maps were found in everyone. Consider tempering this a bit by pointing out that, despite all maps displaying characteristics of typical digit maps, some digit representations are missing. That said, I appreciate the authors including each participant's map in the main paper – very nice to see all the examples!

Line 180: Upon first read, I misinterpreted the statement that "congenital one-handers do not have a missing hand representation". Consider hashing this out a bit more by explicitly stating that they do not develop a cortical representation for the missing body part and hence do not have a hand representation unlike amputees that develop the representation before losing the hand – and hence have a "missing hand representation". This notion is made a bit clearer on lines 266-267; maybe bring forward.

Line 527-529: Given that the map wasn't identified in the patient that reported performing imagined movement as well as the authors' emphasis that the movement was attempted and not imagined (lines 85-86, plus methods), consider adding a bit in the discussion regarding the difference between attempting and imagining a movement – and why they might lead to different activity patterns.

*Reviewer #2 (Recommendations for the authors):*

– I would suggest that the authors perform all analyses including the factor of finger, and that they report finger-specific statistics and tables on all reported analyses. Indeed, looking at the individual maps, it seems that in the patients, fingers are missing. It is also evident that the thumb seems more distant in controls compared to patients. This has to be tested for.

– I would suggest that the authors report % signal change both for the averaged map, for each single finger representation, and for the peak voxels of each finger.

– Prior results on SCI patients have to be described in more detail, also with respect to the used methods (TMS, fMRI).

– To target no 4 in the public review: Remove analyses.

– Conduct analyses on representation order and alignment, map size, and similarity and argue accordingly. Also, the conducted ANOVA is not mentioned in the Methods section, where I indeed wondered why a repeated measure ANOVA was conducted and not a mixed ANOVA with the factor "group". How do the authors want to detect group differences when "group" is not a factor in the ANOVA?

– Methods. It is not clear to me how the mean r was calculated. From what I understand, the authors use the higher r-value as a winner-takes-it-all approach to say which finger wins over the other fingers. When I understand correctly, no statistical threshold is applied here. Therefore, am I correct in assuming that a non-thresholded r map was then transferred to FDR correction?

[Editors' note: further revisions were suggested prior to acceptance, as described below.]

Thank you for resubmitting your work entitled "Finger somatotopy is preserved after tetraplegia but deteriorates over time" for further consideration by *eLife*. Your revised article has been reviewed by 3 reviewers and the evaluation has been overseen by Andrew Pruszynski as the Reviewing Editor and Richard Ivry as the Senior Editor.

The manuscript has been substantially improved and there remain only some small issues that need to be addressed as described below by Reviewers #1 and #2. Please play particular attention to (1) ensuring the abstract is clear and (2) that the finger specific RSA analysis is included in the supplement. We will not send this out to external review again.

*Reviewer #1 (Recommendations for the authors):*

I appreciate the thorough response from the authors and believe that the revised manuscript presents a much more complete and balanced view. My only remaining comment is that although I find the RSA analysis to be appropriate here, I disagree with the authors' claims that it has become the "most common" / "gold standard" approach to investigating inter-finger overlap / somatotopy. And I'm not confident that you'd find consensus for this position among those in the field. As such, consider walking this claim back a bit.

*Reviewer #2 (Recommendations for the authors):*

Overall

I thank the authors for performing additional analyses that improved the quality of manuscript and significantly strengthened the claims they made.

Concept

I thank the authors for adding more background information on nonhuman primates, behavioral and fMRI human studies to argue what (or what not) to expect in their study results. Nevertheless, I still think that the major question posed in the abstract is still not accurate. They write in the abstract:

It is not clear "whether somatotopic representations can be preserved despite alterations in net activity". This is really confusing because in the manuscript, the authors do not show alterations in net activity. In addition, in response to both reviewer 2 and 3, the authors removed the formulation "preserved" from the discussion. Please also remove this from the abstract (at present it is still used 2 times). In addition, as argued by reviewer 1, if attention is one of the mechanisms that triggers the activity, it is not clear whether attention-related net activity should be higher or lower. I would therefore suggest to be precise also in the abstract and write that they investigate "whether somatotopic representations can be activated topographically despite reduced or absent afferent input" (or similar).

Analyses/Results

I thank the authors for conducting an ANOVA including the factors group and finger pair for the RSA, and for conducting finger-specific analyses using percent signal change. It is interesting to see that there is no interaction between group and finger pair, and that individual fingers do not differ in signal change. The authors then say that they conducted individual finger analyses where the BF revealed inconclusive evidence for the RSA analyses. Later in the response letter, they say that those analyses have been included as figure supplement into the manuscript: however, within the material downloaded, I could not find these figures. If the finger pair-specific RSA values are therefore not yet included in the supplemental material, I would suggest adding those because 1. Prior studies hint towards finger-specific differences and researchers would like to see what this study revealed in this respect, 2. The amplitude analyses provide a hint towards differences for D4 that would be interesting to inspect for the RSA results too.

It is good to see that the typicality results did not change when using data of the control group as a basis. This makes this analysis much more convincing. Given the authors argue that it would be good to always use the same data set to compare patients against, in my view, it would be of great benefit for the community if the data and results could be made publicly available (in particular the data taken from the Wesselink study) so that potentially other researchers can compare their own results against those.

*Reviewer #3 (Recommendations for the authors):*

I am satisfied with the revisions.

---

## [Author Response]

Essential revisions:1) The authors need to provide a clearer hypothesis about the mechanistic changes of the sensorimotor maps in the patients. This requires a thorough description of prior results and how these prior results motivated the present study and the analyses conducted. The reviewers emphasize that this will also require additional analyses that bring the present study into alignment with previous work. In this respect, Reviewer 1 and 2 indicate specific figures and analyses that need to be done and presented in a revised manuscript.

We have revised the Introduction accordingly and conducted the analysis suggested by the reviewers. For more details, please see our responses to the specific comments of the reviewers.

2) There is substantial concern that the key "typicality" analysis is not valid. As detailed by Reviewer 2 there are previous studies that have approached this issue but the present manuscript largely ignores this previous literature and uses their own ad hoc approach. The issue of defining typicality needs to be meaningfully (analytically) addressed in reference to previous established approaches.

We respectfully disagree that the typicality measure is not standard, invalid, or inaccurate. The representational structure of the fingers (assessed using RSA) has been shown to depict the invariant organization of S1 better than the exact spatial distribution of finger selective clusters (Ejaz et al., 2015). RSA-based measures are furthermore not prone to some of the problems of the measurements of finger selectivity (e.g., dependence on map thresholds). Indeed, over the past years RSA has become the gold standard to investigate somatotopy of finger representations, both in healthy (e.g. Akselrod et al., 2017; Ejaz et al., 2015; Kieliba et al., 2021; Kolasinski et al., 2016; Sanders et al., 2019) and patient populations (e.g. Dempsey-Jones et al., 2019; Ejaz et al., 2016; Kikkert et al., 2016; Wesselink et al., 2019). Moreover, various papers have been published in *eLife* and elsewhere that used the same RSA-based typicality criteria to assess plasticity in finger representations (Dempsey-Jones et al., 2019; Ejaz et al., 2015; Kieliba et al., 2021; Wesselink et al., 2019). We have added clarification regarding this to Introduction and Methods of the manuscript.

3) There are several analytical points that require attention, either because the choices are unclear or that critical details are not reported .

We thank the reviewers for pointing out this lack of clarity. We have adjusted our text to clarify these choices and added the suggested figures and analysis. Please see our response to the reviewers’ specific comments for further details.

4) Provide a more thorough Discussion, especially in the context of alternative explanations for the present findings. The reviewers all raise important concerns that the present account is very motor-effector dominated and dismisses other potential explanations. In particular, pay close attention to the comments raised by Reviewer #1 in terms of attentional mechanisms and how they may confound the present empirical results. This limitation needs to be addressed either experimentally, or given that possibility is unlikely, explicitly in the Discussion.

We agree that a more balanced speculation of the potential mechanisms underlying the phenomenon of preserved finger mapping would be valuable and have therefore adjusted the Discussion accordingly.

Reviewer #1 (Recommendations for the authors):Abstract: It is unclear what evidence serves the foundation for the authors' claim that rehabilitation approaches would be most promising when applied within the first years after injury. I can see why this might be speculated, but is there any previous work showing that those with in-tact hand representations are more conducive to rehabilitation than those with deteriorated representations?

This is a speculation and we have removed this statement from the abstract in the revised manuscript. Instead, we now state the following:

Revised text Abstract:

“However, over years since SCI the hand representation somatotopy deteriorated, suggesting that somatotopic hand representations are more easily targeted within the first years after SCI.”

Lines 106-116: This section somewhat gives the impression that complete digit maps were found in everyone. Consider tempering this a bit by pointing out that, despite all maps displaying characteristics of typical digit maps, some digit representations are missing. That said, I appreciate the authors including each participant's map in the main paper – very nice to see all the examples!

We agree with the reviewer and we therefore removed the sentence stating that the patient and control maps were qualitatively similar between patients and controls. We instead state that despite all maps displaying aspects of characteristic finger maps, some finger representations were not visible in the thresholded patient and control maps.

Revised text Results:

“Overall, we found aspects of somatotopic finger selectivity in the maps of SCI patients’ hands, in which neighbouring clusters showed selectivity for neighbouring fingers in contralateral S1, similar to those observed in eighteen age-, sex-, and handedness-matched healthy controls. Notably, a characteristic hand map was even found in a patient who suffered complete paralysis and sensory deprivation of the hands (Figure 2, patient map 1; patient S01). Despite most maps (Figure 2, except patient map 3) displaying aspects of characteristic finger selectivity, some finger representations were not visible in the thresholded patient and control maps.”

Line 180: Upon first read, I misinterpreted the statement that "congenital one-handers do not have a missing hand representation". Consider hashing this out a bit more by explicitly stating that they do not develop a cortical representation for the missing body part and hence do not have a hand representation unlike amputees that develop the representation before losing the hand – and hence have a "missing hand representation". This notion is made a bit clearer on lines 266-267; maybe bring forward.

We thank the reviewer for highlighting this to us. We have added the suggested text to these lines in our revised manuscript.

Revised text Results:

“Congenital one-handers are born without a hand and therefore do not have a cortical representation of the missing hand (unlike amputees who develop the representation before losing the hand and hence have a ‘missing hand representation’; Wesselink et al., 2019).”

Line 527-529: Given that the map wasn't identified in the patient that reported performing imagined movement as well as the authors' emphasis that the movement was attempted and not imagined (lines 85-86, plus methods), consider adding a bit in the discussion regarding the difference between attempting and imagining a movement – and why they might lead to different activity patterns.

We thank the reviewer for their suggestion. We are however hesitant to make any strong claims about the putative difference between imagined and attempted movements based on the anecdotal report of this patient. We have revised the manuscript to stress the fact that after the measurement, this patient anecdotally mentioned that he had trouble performing the attempted movements. We are hesitant to speculate about what he might have done instead since we did not include a systematic debriefing procedure to document the patients’ strategies in sufficient detail (e.g., was this patient performing kinaesthetic or visual imagery, or was rather something different). We are aware this is a weakness and have therefore added this debriefing procedure to our follow-up studies.

Revised text Methods:

“We were unable to find a characteristic hand map in one patient who anecdotally reported post hoc that he was unable to perform attempted finger movements.”

Reviewer #2 (Recommendations for the authors):– I would suggest that the authors perform all analyses including the factor of finger, and that they report finger-specific statistics and tables on all reported analyses. Indeed, looking at the individual maps, it seems that in the patients, fingers are missing. It is also evident that the thumb seems more distant in controls compared to patients. This has to be tested for.

With regards to the reviewer’s concern about missing fingers – note that the RSA analysis does not depend on any thresholding (i.e., it works with unthresholded data) and missing fingers due to clusters not surviving thresholding will not impact this analysis. A true missing finger would result in a low typicality score.

Following the reviewer’s suggestion, we have included individual participant inter-finger distance plots for both the controls and patients as Figure 3—figure supplement 2 and Figure 3—figure supplement 3, respectively. We believe this is more readily readable and interpretable than a table containing the 10 inter-finger distance scores for all 32 participants. Instead, these values have been made available online on https://osf.io/e8u95/.

Lastly, we have conducted a geodesic distance analysis to assess whether the finger representations of the SCI patients were aligned differently and/or shifted compared to the control participants. Note however that RSA-based inter-finger distance patterns have been shown to depict the invariant representational structure of fingers in S1 and M1 better than the size, shape, and exact location of the areas activated by finger movements (Ejaz et al., 2015). The geodesic cortical distance was calculated between each participant’s peak finger activation and a reference anchor in the S1 foot cortex. We used the blocked design data for this analysis as the travelling wave paradigm does not lend itself to accurately extract peak finger activation locations due to the consecutive movement task. A robust mixed ANOVA with a within-participants factor for finger (5 levels: thumb, index, middle, ring, and little finger) and a between-participants factor for group (2 levels: controls and patients) showed that, as expected, the geodesic distances were significantly different across fingers (i.e., a main effect for fingers; F_(1,18.81)_ = 4.29, p = 0.05). While the peak finger vertexes were located more lateral for the patient compared to the control group (i.e., a main effect for group; F_(4,13.93)_ = 85.66, p < 0.001), the geodesic distance pattern across the fingers was not significantly different across groups (i.e., no significant finger*group interaction; F_(4,13.93)_ = 0.91, p = 0.48). When testing for group differences per finger, the BF only revealed inconclusive evidence (BF > 0.34 and < 1.88; note that we could not run a Bayesian ANOVA due to normality violations). We have added this analysis to Appendix 1.

Revised text Methods:

“We further compared activity levels in finger-specific ROIs in S1 between groups and conducted a geodesic distance analysis to assess whether the finger representations of SCI patients were aligned differently and/or shifted compared to the control group (see Appendix 1).”

Please see Appendix 1, “Geodesic distance analysis” section for the revised text.

– I would suggest that the authors report % signal change both for the averaged map, for each single finger representation, and for the peak voxels of each finger.

We have adjusted our analysis, results, and figures in our main manuscript such that we now report % signal change rather than averaged z-standardised BOLD activity in the revised main manuscript.

– Prior results on SCI patients have to be described in more detail, also with respect to the used methods (TMS, fMRI).

We revised our Introduction accordingly.

– To target no 4 in the public review: Remove analyses.

We respectfully disagree with the reviewer on removing this analysis. We elaborated on the validity of our typicality analysis in response to public review comment 2.6.

– Conduct analyses on representation order and alignment, map size, and similarity and argue accordingly. Also, the conducted ANOVA is not mentioned in the Methods section, where I indeed wondered why a repeated measure ANOVA was conducted and not a mixed ANOVA with the factor "group". How do the authors want to detect group differences when "group" is not a factor in the ANOVA?

We have removed the sentence saying that ‘qualitatively the position, order of finger preference, and extent of the hand maps were generally similar between patients and controls’. We indeed do not support this claim with quantitative analysis. As per the suggestion of reviewer 1, we now state that despite all maps (except patient map 3) displaying aspects of characteristic finger maps, some finger representations were not visible in the thresholded patient and control maps.

As also indicated in our response to comment 2.11, to assess whether the maps of the SCI patients were aligned differently and/or shifted compared to the control participants, we extracted the geodesic distance per finger on a normalised cortical surface from each participant’s peak activated vertex to a reference vertex placed in the S1 foot cortex.

The ANOVA we conducted was an ANOVA with a within-participants factor for finger pair (3 levels: same, neighbouring and non-neighbouring finger pairs) and a between-participants factor for group (2 levels: controls and patients). We clarified this in the revised manuscript.

Revised text Methods:

“We tested whether the somatotopic relationship in the DOCs was different in controls and patients using a robust mixed ANOVA with a within-participants factor for finger pair (3 levels: same, neighbouring and non-neighbouring finger pairs) and a between-participants factor for group (2 levels: controls and patients).”

Revised text Results:

“Overall, split-half consistency was not significantly different between patients and controls, as tested using a robust mixed ANOVA (see Figure 2C; F_(1,20.32)_ = 0.51, p = 0.48).”

– Methods. It is not clear to me how the mean r was calculated. From what I understand, the authors use the higher r-value as a winner-takes-it-all approach to say which finger wins over the other fingers. When I understand correctly, no statistical threshold is applied here. Therefore, am I correct in assuming that a non-thresholded r map was then transferred to FDR correction?

We thank the reviewer for pointing out the lack of clarity. After calculating the r-values using the cross-correlation approach, the r-values were normalized using a Fisher r-to-z transformation. Following this, we used a winner-take-all approach to assign each voxel to one finger/lag exclusively based on the maximum z-value. We then calculated the threshold using the FDR q < 0.05 criterion based on winner-take-all finger/lag z-valued maps in native space (3D) and applied this threshold to the winner-take-all maps on the surface. We have clarified this in the Methods.

Revised text Methods:

“This resulted in 25 r-values per voxel per run that were normalised using a Fisher r-to-z transformation. […] Thresholding was applied to the winner-take-all finger-specific maps and lag-specific maps on the cortical surface using a false discovery criterion q < 0.05 based on the native (3D) values.”

References:

Adams RA, Shipp S, Friston KJ. 2013. Predictions not commands: Active inference in the motor system. Brain Struct Funct. doi:10.1007/s00429-012-0475-5

Akselrod M, Martuzzi R, Serino A, van der Zwaag W, Gassert R, Blanke O. 2017. Anatomical and functional properties of the foot and leg representation in areas 3b, 1 and 2 of primary somatosensory cortex in humans: A 7T fMRI study. Neuroimage 159:473–487. doi:10.1016/j.neuroimage.2017.06.021

Armenta Salas M, Bashford L, Kellis S, Jafari M, Jo H, Kramer D, Shanfield K, Pejsa K, Lee B, Liu CY, Andersen RA. 2018. Proprioceptive and cutaneous sensations in humans elicited by intracortical microstimulation. *eLife* 7:1–11. doi:10.7554/*eLife*.32904

Berlot E, Prichard G, O’Reilly J, Ejaz N, Diedrichsen J. 2019. Ipsilateral finger representations in the sensorimotor cortex are driven by active movement processes, not passive sensory input. J Neurophysiol 121:418–426. doi:10.1152/jn.00439.2018

Besle J, Sánchez-Panchuelo R-M, Bowtell R, Francis S, Schluppeck D. 2013. Single-subject fMRI mapping at 7 T of the representation of fingertips in S1: a comparison of event-related and phase-encoding designs. J Neurophysiol 109:2293–305. doi:10.1152/jn.00499.2012

Bruurmijn MLCM, Pereboom IPL, Vansteensel MJ, Raemaekers MAH, Ramsey NF. 2017. Preservation of hand movement representation in the sensorimotor areas of amputees. Brain 140:3166–3178. doi:10.1093/brain/awx274

Burman DD, Lie-Nemeth T, Brandfonbrener AG, Parisi T, Meyer JR. 2009. Altered finger representations in sensorimotor cortex of musicians with focal dystonia: Precentral cortex. Brain Imaging Behav. doi:10.1007/s11682-008-9046-z

Chand P, Jain N. 2015. Intracortical and thalamocortical connections of the hand and face representations in somatosensory area 3b of macaque monkeys and effects of chronic spinal cord injuries. J Neurosci 35:13475–13486. doi:10.1523/JNEUROSCI.2069-15.2015

Chen LM, Qi H-X, Kaas JH. 2012. Dynamic Reorganization of Digit Representations in Somatosensory Cortex of Nonhuman Primates after Spinal Cord Injury. J Neurosci 32:14649–14663. doi:10.1523/JNEUROSCI.1841-12.2012

Cramer SC, Lastra L, Lacourse MG, Cohen MJ. 2005. Brain motor system function after chronic, complete spinal cord injury. Brain 128:2941–2950. doi:10.1093/brain/awh648

Da Costa S, Saenz M, Clarke S, van der Zwaag W. 2014. Tonotopic Gradients in Human Primary Auditory Cortex: Concurring Evidence From High-Resolution 7 T and 3 T fMRI. Brain Topogr 28:66–69. doi:10.1007/s10548-014-0388-0

Da Rocha Amaral S, Sanchez Panchuelo RM, Francis S. 2020. A Data-Driven Multi-scale Technique for fMRI Mapping of the Human Somatosensory Cortex. Brain Topogr 33:22–36. doi:10.1007/s10548-019-00728-6

Delhaye BP, Long KH, Bensmaia SJ. 2019. Neural basis of touch and proprioception in primate cortex. Compr Physiol 8:1575–1602. doi:10.1002/cphy.c170033

Dempsey-Jones H, Wesselink DB, Friedman J, Makin TR. 2019. Organized Toe Maps in Extreme Foot Users. Cell Rep 28:2748-2756.e4. doi:10.1016/j.celrep.2019.08.027

Deyoe EA, Carman GJ, Bandettini P, Glickman S, Wieser J, Cox R, Miller D, Neitz J. 1996. Mapping striate and extrastriate visual areas in human cerebral cortex. Proc Natl Acad Sci U S A 93:2382–2386. doi:10.1073/pnas.93.6.2382

Ejaz N, Hamada M, Diedrichsen J. 2015. Hand use predicts the structure of representations in sensorimotor cortex. Nat Neurosci 18:1034–1040.

Ejaz N, Sadnicka A, Wiestler T, Butler K, Edwards M, Diedrichsen J. 2016. Finger representations in sensorimotor cortex are not disrupted in musicians’ dystoniaSociety for Neuroscience Annual Meeting, San Diego, USA.

Engel SA, Glover GG, Wandell BA. 1997. Retintopic organization in human visual cortex and the spatial precision of functional MRI. Cereb Cortex 7:181–192. doi:10.1093/cercor/7.2.181

Fassett HJ, Turco C V., El-Sayes J, Nelson AJ. 2018. Alterations in motor cortical representation of muscles following incomplete spinal cord injury in humans. Brain Sci. doi:10.3390/brainsci8120225

Fifer MS, McMullen DP, Thomas TM, Osborn LE, Nickl RW, Candrea DN, Pohlmeyer EA, Thompson MC, Anaya M, Schellekens W, Ramsey NF, Bensmaia SJ, Anderson WS, Wester BA, Crone NE, Celnik PA, Cantarero GL, Tenore F V. 2020. Intracortical Microstimulation Elicits Human Fingertip Sensations. medRxiv 2020.05.29.20117374.

Flesher SN, Collinger JL, Foldes ST, Weiss JM, Downey JE, Tyler-Kabara EC, Bensmaia SJ, Schwartz AB, Boninger ML, Gaunt RA. 2016. Intracortical microstimulation of human somatosensory cortex. Sci Transl Medicne 1–11. doi:10.1126/scitranslmed.aaf8083

Florence SL. 1998. Large-Scale Sprouting of Cortical Connections After Peripheral Injury in Adult Macaque Monkeys. Science (80- ) 282:1117–1121. doi:10.1126/science.282.5391.1117

Freund P, Rothwell J, Craggs M, Thompson AJ, Bestmann S. 2011a. Corticomotor representation to a human forearm muscle changes following cervical spinal cord injury. Eur J Neurosci 34:1839–1846. doi:10.1111/j.1460-9568.2011.07895.x

Freund P, Schmidlin E, Wannier T, Bloch J, Mir A, Schwab ME, Rouiller EM. 2006. Nogo-A-specific antibody treatment enhances sprouting and functional recovery after cervical lesion in adult primates. Nat Med. doi:10.1038/nm1436

Freund P, Weiskopf N, Ward NS, Hutton C, Gall A, Ciccarelli O, Craggs M, Friston K, Thompson AJ. 2011b. Disability, atrophy and cortical reorganization following spinal cord injury. Brain 134:1610–1622. doi:10.1093/brain/awr093

Halder P, Kambi N, Chand P, Jain N. 2018. Altered Expression of Reorganized Inputs as They Ascend From the Cuneate Nucleus to Cortical Area 3b in Monkeys With Long-Term Spinal Cord Injuries. Cereb Cortex 28:3922–3938. doi:10.1093/cercor/bhx256

Henderson LA, Gustin SM, Macey PM, Wrigley PJ, Siddall PJ. 2011. Functional reorganization of the brain in humans following spinal cord injury: Evidence for underlying changes in cortical anatomy. J Neurosci 31:2630–2637. doi:10.1523/JNEUROSCI.2717-10.2011

Hotz-Boendermaker S, Funk M, Summers P, Brugger P, Hepp-Reymond MC, Curt A, Kollias SS. 2008. Preservation of motor programs in paraplegics as demonstrated by attempted and imagined foot movements. Neuroimage 39:383–394. doi:10.1016/j.neuroimage.2007.07.065

Huber E, Lachappelle P, Sutter R, Curt A, Freund P. 2017. Are midsagittal tissue bridges predictive of outcome after cervical spinal cord injury? Ann Neurol 81:740–748. doi:10.1002/ana.24932

Jain N, Qi H-X, Collins CE, Kaas JH. 2008. Large-scale reorganization in the somatosensory cortex and thalamus after sensory loss in macaque monkeys. J Neurosci 28:11042–11060. doi:10.1523/JNEUROSCI.2334-08.2008

Jutzeler CR, Freund P, Huber E, Curt A, Kramer JLK. 2015. Neuropathic pain and functional reorganization in the primary sensorimotor cortex after spinal cord injury. J Pain 16:1256–1267. doi:10.1016/j.jpain.2015.08.008

Kambi N, Halder P, Rajan R, Arora V, Chand P, Arora M, Jain N. 2014. Large-scale reorganization of the somatosensory cortex following spinal cord injuries is due to brainstem plasticity. Nat Commun 5:3602. doi:10.1038/ncomms4602

Kieliba P, Clode D, Maimon-mor RO, Makin TR. 2021. Robotic hand augmentation drives changes in neural body representation. Sci Robot 6:1–14. doi:10.1126/scirobotics.abd7935

Kikkert S, Kolasinski J, Jbabdi S, Tracey I, Beckmann CF, Johansen-Berg H, Makin TR. 2016. Revealing the neural fingerprints of a missing hand. *eLife* 5:e15292. doi:10.7554/*eLife*.15292

Kokotilo K, Eng J, Curt A. 2009. Reorganization and preservation of motor control of the brain in spinal cord injury: a systematic review. J Neurotrauma 26:2113–2126. doi:10.1089/neu.2008.0688.Reorganization

Kokotilo KJ, Eng JJ, Curt A. 2009. Reorganization and preservation of motor control of the brain in spinal cord injury: A systematic review. J Neurotrauma 26:2113–2126. doi:10.1089/neu.2008.0688

Kolasinski J., Makin TR, Jbabdi S, Clare S, Stagg CJ, Johansen-Berg H. 2016. Investigating the stability of fine-grain digit somatotopy in individual human participants. J Neurosci 36:1113–1127.

Kolasinski James, Makin TR, Logan JP, Jbabdi S, Clare S, Stagg CJ, Johansen-Berg H. 2016. Perceptually relevant remapping of human somatotopy in 24 hours. *eLife* 5:1–15. doi:10.7554/*eLife*.17280

Kriegeskorte N, Mur M, Bandettini P. 2008. Representational similarity analysis – connecting the branches of systems neuroscience. Front Syst Neurosci 2:1–28. doi:10.3389/neuro.06.004.2008

Kuehn E, Haggard P, Villringer A, Pleger B, Sereno MI. 2018. Visually-driven maps in area 3b. J Neurosci 38:0491–17. doi:10.1523/JNEUROSCI.0491-17.2017

Levy WJ, Amassian VE, Traad M, Cadwell J. 1990. Focal magnetic coil stimulation reveals motor cortical system reorganized in humans after traumatic quadriplegia. Brain Res. doi:10.1016/0006-8993(90)90738-W

London BM, Miller LE. 2013. Responses of somatosensory area 2 neurons to actively and passively generated limb movements. J Neurophysiol 109:1505–13. doi:10.1152/jn.00372.2012

Mancini F, Haggard P, Iannetti GD, Longo MR, Sereno MI. 2012. Fine-grained nociceptive maps in primary somatosensory cortex. J Neurosci 32:17155–17162. doi:10.1523/JNEUROSCI.3059-12.2012

Nili H, Wingfield C, Walther A, Su L, Marslen-Wilson W, Kriegeskorte N. 2014. A Toolbox for Representational Similarity Analysis. PLoS Comput Biol 10. doi:10.1371/journal.pcbi.1003553

Pfyffer D, Huber E, Sutter R, Curt A, Freund P. 2019. Tissue bridges predict recovery after traumatic and ischemic thoracic spinal cord injury. Neurology 93:e1550–e1560. doi:10.1212/WNL.0000000000008318

Pfyffer D, Vallotton K, Curt A, Freund P. 2021. Predictive Value of Midsagittal Tissue Bridges on Functional Recovery After Spinal Cord Injury. Neurorehabil Neural Repair 35:33–43. doi:10.1177/1545968320971787

Puckett AM, Bollmann S, Barth M, Cunnington R. 2017. Measuring the effects of attention to individual fingertips in somatosensory cortex using ultra-high field (7T) fMRI. Neuroimage 161:179–187. doi:10.1016/j.neuroimage.2017.08.014

Puckett AM, Bollmann S, Junday K, Barth M, Cunnington R. 2020. Bayesian population receptive field modeling in human somatosensory cortex. Neuroimage 208:116465. doi:10.1016/j.neuroimage.2019.116465

Sanders Z-B, Wesselink DB, Dempsey-Jones H, Makin TR. 2019. Similar somatotopy for active and passive digit representation in primary somatosensory cortex. bioRxiv 1–36. doi:10.1101/754648

Sereno MI, Dale AM, Reppas JB, Kwong KK, Belliveau JW, Brady TJ, Rosen BR, Tootell RBH. 1995. Borders of multiple visual areas in humans revealed by functional magnetic resonance imaging. Science (80- ) 268:889–893. doi:10.1126/science.7754376

Solstrand Dahlberg L, Becerra L, Borsook D, Linnman C. 2018. Brain changes after spinal cord injury, a quantitative meta-analysis and review. Neurosci Biobehav Rev 90:272–293. doi:10.1016/j.neubiorev.2018.04.018

Streletz LJ, Belevich JKS, Jones SM, Bhushan A, Shah SH, Herbison GJ. 1995. Transcranial magnetic stimulation: Cortical motor maps in acute spinal cord injury. Brain Topogr. doi:10.1007/BF01202383

Talavage TM, Sereno MI, Melcher JR, Ledden PJ, Rosen BR, Dale AM. 2004. Tonotopic Organization in Human Auditory Cortex Revealed by Progressions of Frequency Sensitivity. J Neurophysiol 91:1282–1296. doi:10.1152/jn.01125.2002

Taub E, Elbert T, Candia V, Rockstroh B, Rau H, Pantev C, Altenmüller E, Sterr A. 1998. Alteration of digital representations in somatosensory cortex in focal hand dystonia. Neuroreport.

Topka H, Cohen LG, Cole RA, Hallett M. 1991. Reorganization of corticospinal pathways following spinal cord injury. Neurology. doi:10.1212/wnl.41.8.1276

Urbin MA, Royston DA, Weber DJ, Boninger ML, Collinger JL. 2019. What is the functional relevance of reorganization in primary motor cortex after spinal cord injury? Neurobiol Dis 121:286–295. doi:10.1016/j.nbd.2018.09.009

Vallotton K, Huber E, Sutter R, Curt A, Hupp M, Freund P. 2019. Width and neurophysiologic properties of tissue bridges predict recovery after cervical injury. Neurology 92:E2793–E2802. doi:10.1212/WNL.0000000000007642

Wesselink DB, Maimon-Mor R. 2017. Github.

Wesselink DB, van den Heiligenberg FM, Ejaz N, Dempsey-Jones H, Cardinali L, Tarall-Jozwiak A, Diedrichsen J, Makin TR. 2019. Obtaining and maintaining cortical hand representation as evidenced from acquired and congenital handlessness. *eLife* 8:1–19. doi:10.7554/*eLife*.37227

Wrigley PJ, Press SR, Gustin SM, Macefield VG, Gandevia SC, Cousins MJ, Middleton JW, Henderson LA, Siddall PJ. 2009. Neuropathic pain and primary somatosensory cortex reorganization following spinal cord injury. Pain 141:52–59. doi:10.1016/j.pain.2008.10.007

Wrigley PJ, Siddall PJ, Gustin SM. 2018. New evidence for preserved somatosensory pathways in complete spinal cord injury: A fMRI study. Hum Brain Mapp 39:588–598. doi:10.1002/hbm.23868

Zeharia N, Hertz U, Flash T, Amedi A. 2015. New whole-body sensory-motor gradients revealed using phase-locked analysis and verified using multivoxel pattern analysis and functional connectivity. J Neurosci 35:2845–2859. doi:10.1523/Jneurosci.4246-14.2015

[Editors' note: further revisions were suggested prior to acceptance, as described below.]

The manuscript has been substantially improved and there remain only some small issues that need to be addressed as described below by Reviewers #1 and #2. Please play particular attention to (1) ensuring the abstract is clear and (2) that the finger specific RSA analysis is included in the supplement. We will not send this out to external review again.Reviewer #1 (Recommendations for the authors):I appreciate the thorough response from the authors and believe that the revised manuscript presents a much more complete and balanced view. My only remaining comment is that although I find the RSA analysis to be appropriate here, I disagree with the authors' claims that it has become the "most common" / "gold standard" approach to investigating inter-finger overlap / somatotopy. And I'm not confident that you'd find consensus for this position among those in the field. As such, consider walking this claim back a bit.Reviewer #2 (Recommendations for the authors):OverallI thank the authors for performing additional analyses that improved the quality of manuscript and significantly strengthened the claims they made.ConceptI thank the authors for adding more background information on nonhuman primates, behavioral and fMRI human studies to argue what (or what not) to expect in their study results. Nevertheless, I still think that the major question posed in the abstract is still not accurate. They write in the abstract:It is not clear "whether somatotopic representations can be preserved despite alterations in net activity". This is really confusing because in the manuscript, the authors do not show alterations in net activity. In addition, in response to both reviewer 2 and 3, the authors removed the formulation "preserved" from the discussion. Please also remove this from the abstract (at present it is still used 2 times). In addition, as argued by reviewer 1, if attention is one of the mechanisms that triggers the activity, it is not clear whether attention-related net activity should be higher or lower. I would therefore suggest to be precise also in the abstract and write that they investigate "whether somatotopic representations can be activated topographically despite reduced or absent afferent input" (or similar).Analyses/ResultsI thank the authors for conducting an ANOVA including the factors group and finger pair for the RSA, and for conducting finger-specific analyses using percent signal change. It is interesting to see that there is no interaction between group and finger pair, and that individual fingers do not differ in signal change. The authors then say that they conducted individual finger analyses where the BF revealed inconclusive evidence for the RSA analyses. Later in the response letter, they say that those analyses have been included as figure supplement into the manuscript: however, within the material downloaded, I could not find these figures. If the finger pair-specific RSA values are therefore not yet included in the supplemental material, I would suggest adding those because 1. Prior studies hint towards finger-specific differences and researchers would like to see what this study revealed in this respect, 2. The amplitude analyses provide a hint towards differences for D4 that would be interesting to inspect for the RSA results too.It is good to see that the typicality results did not change when using data of the control group as a basis. This makes this analysis much more convincing. Given the authors argue that it would be good to always use the same data set to compare patients against, in my view, it would be of great benefit for the community if the data and results could be made publicly available (in particular the data taken from the Wesselink study) so that potentially other researchers can compare their own results against those.

Specifically:

– We removed our statement about RSA being the golden standard to investigate inter-finger overlap, as per the suggestion of Reviewer 1.

Revised text Methods:

“We estimated inter-finger overlap using RSA. Note that it is also possible to estimate somatotopic overlap from travelling wave data using an iterated Multigrid Priors (iMGP) method and population receptive field modelling (Da Rocha Amaral et al., 2020; Puckett et al., 2020).”

– We changed the terminology in our abstract according as per the suggestion of Reviewer 2.

Revised text Abstract:

“However, little is known about whether somatotopically-specific representations can be activated despite reduced or absent afferent hand inputs. […] We found that somatotopic hand representations can be activated through attempted finger movements in absence of sensory and motor hand functioning, and no spared spinal tissue bridges.

– We added a figure supplement to visualise inter-finger distances across finger pairs for controls and SCI patients, as per the suggestion of Reviewer 2.

Revised text Results:

“We then tested whether the inter-finger distances were different across finger pairs between controls and SCI patients using a robust mixed ANOVA with a within-participants factor for finger pair (10 levels) and a between-participants factor for group (2 levels: controls and patients; Figure 3—figure supplement 2).”

– The canonical RDM used for analysis has already been made publicly available on https://osf.io/e8u95/. In the revised manuscript we refer readers to the publicly available congenital amputees dataset (Wesselink et al., 2019) in the Methods section, according to the suggestion of Reviewer 2.

Revised text Methods:

“Controls’ and SCI patients’ typicality scores were compared to those of a group of individuals with congenital hand malformation (n = 13), hereafter one-handers, obtained in another study (data publicly available on https://osf.io/gmvua/; Wesselink et al., 2019).”